# Stress drives plasticity in leaf ageing transcriptional dynamics in *Arabidopsis thaliana*

**Joseph Swift** [1] ✉, **Xuelin Wu** [1,2], **Jiaying Xu**[1], **Carl Procko** [1,2], **Tanvi Jain**[1], **Natanella Illouz-Eliaz** [1], **Joseph R. Nery** [3], **Joanne Chory** [1,2] & **Joseph R. Ecker** [1,2,3] ✉

Leaf development is dynamic, enabling plants to modulate their growth in response to environmental cues. Under drought conditions, for instance, the model plant *Arabidopsis thaliana* restricts leaf growth to conserve water, a strategy that enhances water-use efficiency. While this 'stress avoidance' response is well described physiologically, the underlying transcriptional changes that drive such developmental plasticity remain poorly understood. We investigated the transcriptional basis of how drought stress reshapes *Arabidopsis* leaf development. We profiled 1,226 leaves at various developmental stages and levels of drought stress, and generated a single-nucleus transcriptome atlas comprising ~1 million individual nuclei. We found that drought stress advances transcriptional programmes associated with leaf ageing in a dose-dependent manner, particularly within the mesophyll. These transcriptional shifts scale with stress intensity and correlate with reduced shoot growth, indicating that mesophyll-specific transcriptional changes underlie drought-induced restriction in leaf growth. Overexpression of *FERRIC REDUCTION OXIDASE 6* (*FRO6*) in the mesophyll was sufficient to partially restore leaf growth under drought conditions. Our findings demonstrate how gene expression is reshaped by environmental cues to ensure that shoot architecture is adaptive to stress severity.

Plant leaf growth progresses through ordered stages of development. Yet, unlike mammalian development, it exhibits a remarkable degree of plasticity, responding dynamically to environmental cues. For instance, upon encountering drought stress, the model plant *Arabidopsis thaliana* limits the size of growing leaves and induces senescence in older ones[1–3]. Referred to as the 'stress avoidance' strategy[4], this phenotypic change leads to a compact plant stature that is more water-use efficient and helps plants survive when drought conditions arise[4,5]. The gene expression responses that drive such plasticity are not well understood. Since the lifespan of a leaf is governed by the sequential expression of

genes involved in cellular proliferation, expansion and senescence[6], changing the timing or induction levels of such genes may be one way in which drought signalling can impact the course of leaf development. Indeed, drought stress has been shown to downregulate genes associated with leaf growth rate[7], as well as to induce cell cycle exit[8]. However, a complete picture of how the transcriptional events that drive leaf development are reshaped by drought stress remains incomplete.

To address this, we characterized the transcriptional responses among *Arabidopsis* leaf cell types as they aged, and explored how their expression dynamics changed as drought intensified. Across cell

[1]Plant Biology Laboratory, The Salk Institute for Biological Studies, La Jolla, CA, USA. [2]Howard Hughes Medical Institute, The Salk Institute for Biological Studies, La Jolla, CA, USA. [3]Genomic Analysis Laboratory, The Salk Institute for Biological Studies, La Jolla, CA, USA. ✉e-mail: jswift@salk.edu; ecker@salk.edu

**Fig. 1 | Transcriptional states of *Arabidopsis* cell types change during leaf ageing. a**, Images of 29-day-old (day 1 of the experiment) (left) and 37-day-old (day 9 of the experiment) (right) *Arabidopsis* rosettes with leaf stage indicated (scale bar, 1 cm). Below, leaf area of each leaf stage sampled at the beginning (day 1) (left) and end (day 9) (right) of the time course is shown (bars indicates mean, *n* = 3 leaves per stage). **b**, A total of 647 individual leaves spanning 15 leaf developmental stages across the 9-day time course were profiled at single-nucleus resolution using sci-RNA-seq3. **c**, Schematic illustrating the leaf stages and time points sampled across the experiment. **d**, A leaf transcriptional atlas was assembled from 264,183 nuclei, with nine leaf cell types identified using established marker genes. **e**, Epidermal nuclei subclustered and coloured by the leaf stage from which they were sourced. **f**, Expression patterns of *CYCD1;1*, *CYCB1;2*, *PME5*, *RD29A* and *KMD1* within the epidermal subcluster. **g**, Pseudobulked expression profiles of *CYCD1;1* and *KMD1* in the epidermis subcluster, arranged by leaf stage and day of sampling. **h**, Normalized expression profiles of *CYCD1;1*, *CYCB1;2*, *PME5*, *RD29A* and *KMD1* in the epidermis subcluster across leaf age. Curve shows quadratic model fit (solid lines) with 95% CI indicated by shaded area. **i**, The *z*-score normalized expression trends of 115 genes upregulated specifically within the epidermis subcluster during leaf maturation. Curves show quadratic model fit (solid lines), with 99% CIs indicated by shaded areas. Select gene names are shown.

types, we found that drought stress primarily promoted the expression of genes associated with leaf maturation and ageing, coinciding with changes in hormone signalling related to leaf development. Furthermore, we determined that development-associated genes adjusted their expression in a dose-dependent manner, indicating how transcriptional signals help to align shoot growth with stress intensity. Our results led us to identify *FERRIC REDUCTION OXIDASE 6* (*FRO6*) within the mesophyll cell type and, by perturbing its cell-type-specific expression, found it to be a regulator of *Arabidopsis* shoot size under drought stress.

## Dynamic transcriptional cell states underlie leaf ageing

Leaf ageing is defined by the progression of the physiology of a leaf over time, from its initial emergence through to the last stages of its lifespan[9]. We sought to understand how cell-type-specific gene expression patterns change as leaves age. Because *Arabidopsis* rosettes naturally contain leaves at different stages of development, we leveraged this internal developmental gradient and sampled 15 leaf development stages (L1–L15) (Fig. 1a). The youngest leaves sampled were approximately 20 mm², a size typical of a young leaf undergoing expansion[10], while the largest fully expanded leaves reached above

235 mm² (Fig. 1a). To capture transcriptional changes as each leaf stage grew over time, we sampled each leaf stage over a 9-day time course. Thus, by sampling leaves that varied both in their developmental stage, as well as across time, we collected 647 leaves representing a range of different ages within the lifespan of a leaf (Fig. 1b,c). To profile the transcriptomes of individual cell types within each leaf, we used sci-RNA-seq3, a plate-based single-nucleus RNA sequencing method that uses three rounds of molecular indexing[11]. Crucially, we applied the first round of indexing (the reverse transcription step) to uniquely barcode nuclei from each of the different 647 leaves sampled (Fig. 1b). This allowed us to trace transcripts within each sequenced nucleus back to its leaf of origin, enabling high-throughput, leaf-resolved, single-nucleus transcriptomic profiling. Using this approach, we constructed a single-nucleus atlas of the *Arabidopsis* leaf comprising 264,183 nuclei (Fig. 1d and Supplementary Table 1). On the basis of established cell identity markers, we annotated nine distinct leaf cell types within the atlas (Extended Data Fig. 1 and Supplementary Table 2). This atlas can be explored through an interactive browser (Extended Data Fig. 2).

Within this atlas, we first examined the epidermal cell type, as it is known to play a key role in regulating leaf development[12]. When we subclustered epidermal nuclei, they grouped by the leaf developmental

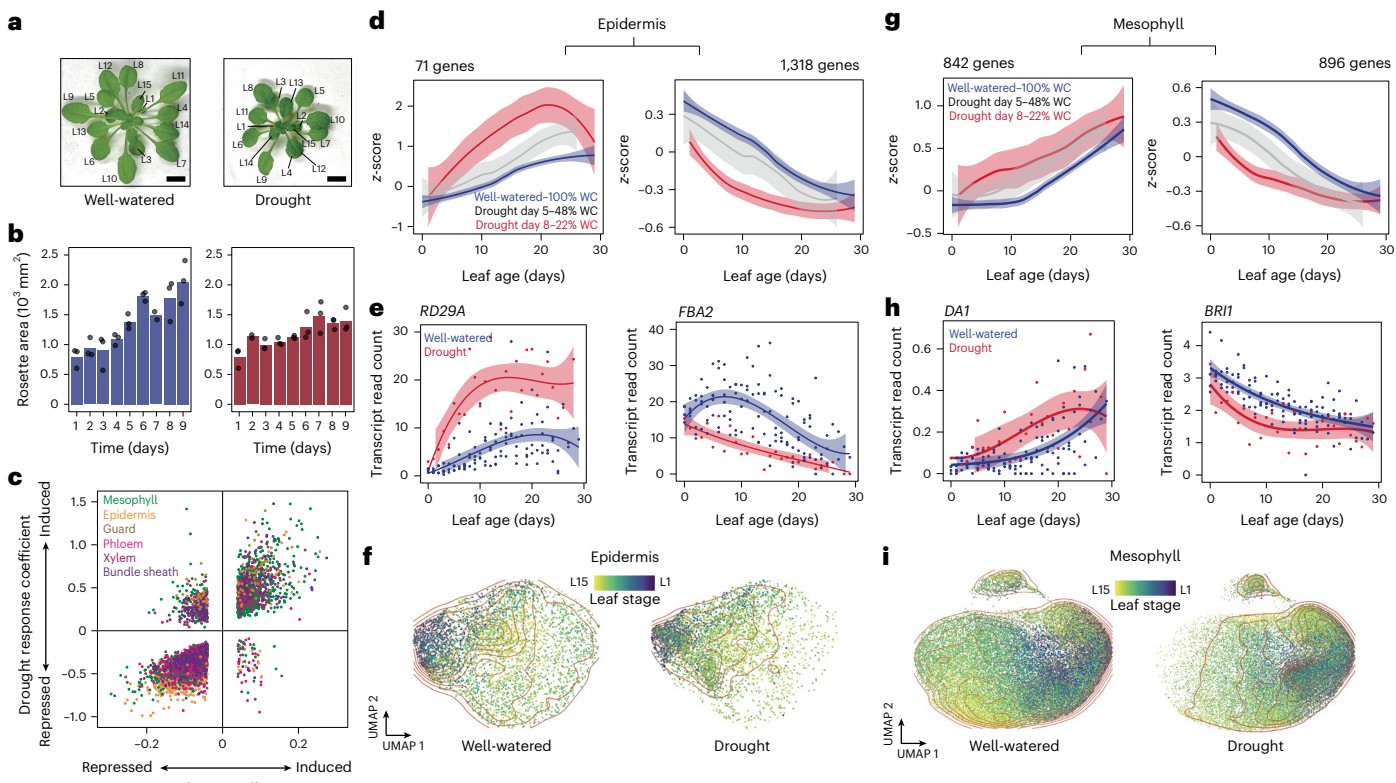

**Fig. 2 | Drought stress promotes leaf-ageing transcriptional dynamics.**
**a**, Images of 37-day-old *Arabidopsis* rosettes grown under well-watered conditions or subjected to drought for 9 days (leaf stage indicated; scale bar, 1 cm). **b**, Shoot area of rosettes grown under each treatment during the 9-day time course (bar indicates mean, ANCOVA test between conditions $P = 3.4 \times 10^{-3}$, $n = 3$ individual rosettes, except day 7 (well-watered) and day 2 (drought), where $n = 2$). **c**, Induction or repression of leaf-ageing-associated genes (each dot represents a gene) that were differentially expressed in response to drought stress (colour indicates cell type). **d**, Scaled expression of genes induced or repressed during leaf ageing in the epidermal cell type under three levels of WC within the pot used for the plant (solid line displays quadratic model fit to expression data, shaded areas indicate 99% CI). **e**, *RD29A* and *FBA2* expression within the epidermis cell type (dots) under well-watered conditions and

drought conditions (22–28% WC, solid line fit using quadratic model, shaded areas indicate 95% CI). **f**, Epidermal nuclei from well-watered or drought conditions subclustered and coloured by their respective leaf stage (red contour lines indicate equal point density; both UMAPs show 4,812 nuclei). **g**, Scaled expression of genes induced or repressed during leaf ageing in the mesophyll cell type under three levels of WC (solid line displays quadratic model fit to scaled average expression, shaded areas indicate 99% CI). **h**, *DA1* and *BRI1* expression within the mesophyll cell type of leaves (dots) under well-watered and drought conditions (22–28% WC, solid line fit using quadratic model, shade areas indicate 95% CI). **i**, Mesophyll nuclei sourced from well-watered or drought conditions were subclustered and coloured by their respective leaf stage (red contour lines indicate equal point density; both UMAPs show 30,270 nuclei).

stage they were isolated from (Fig. 1e and Extended Data Fig. 3). Among epidermal nuclei sourced from younger leaves (L15–L10), consistent with active endoreduplication, we observed high expression of *CYCLIN D1;1* (*CYCD1;1*)[13]. We also saw higher expression of cell cycle regulator *CYCLIN B1;2* (*CYCB1;2*), as well as cell wall loosening and expansion related gene *PECTIN METHYLESTERASE 5* (*PME5*)[14] (Fig. 1f). In contrast, nuclei from older leaves (L5–L1) expressed genes associated with senescence, such as the cytokinin (CK) signalling repressor *KISS ME DEADLY 1* (*KMD1*)[15] and the stress-responsive gene *RESPONSE TO DESICCATION 29 A* (*RD29A*)[16] (Fig. 1g). These patterns indicate that our atlas captured the cell-type-specific transcriptional signatures spanning from expansion to late maturity developmental stages, thus extending previous bulk RNA-seq observations[6] to single-nuclei resolution.

Within our atlas, we sought to identify genome-wide expression trends associated with leaf ageing. To achieve this, we applied a linear model to pseudobulked transcriptional profiles of each cell type. For the expression of a gene to be associated with leaf ageing, we required its expression to change both across leaf developmental stage (from L15 to L1) as well as across real time (days 1–9) (Extended Data Fig. 4). Because our model required genes to change expression both across leaf stage and real time, by design it excluded genes associated with discrete developmental transitions, such as the juvenile-to-adult shift or heteroblasty.

For example, a gene that met this criteria was *KMD1* in the epidermis. Here its expression increased both as leaf stage and as time progressed (Fig. 1g).

Using this modelling approach, we identified hundreds of cell-type-specific genes whose expression patterns significantly increased or decreased as leaves aged ($P_{adj} < 0.01$, linear model; Supplementary Table 3), including genes known to regulate leaf development (Extended Data Fig. 3). For instance, we found 115 genes that were specifically upregulated in the epidermis cell type as leaves aged, holding enriched gene ontology (GO) terms such as 'response to biotic stress' and 'response to auxin stimulus' ($P_{adj} < 8.8 \times 10^{-3}$) (Fig. 1h). Among these were the leaf senescence regulators *INDOLE-3-ACETIC ACID INDUCIBLE 29* (*IAA29*) and *WRKY DNA-BINDING PROTEIN 38* (*WRKY38*)[17]. Additionally, we found 733 genes that were specifically downregulated in the epidermis as leaves aged, including genes known to positively influence leaf size, such as *TEOSINTE BRANCHED 1/CYCLOIDEA/PROLIFERATING CELL NUCLEAR ANTIGEN FACTOR 2;3;4* (*TCP2;3;4*), *GA INSENSITIVE* (*GAI*) and *CYCB1;1*[18] (Extended Data Fig. 3).

## Drought stress advances leaf-ageing transcriptional dynamics

When drought conditions arise, *Arabidopsis* restricts leaf size as a form of stress avoidance[4]. As part of this response, cellular expansion in the

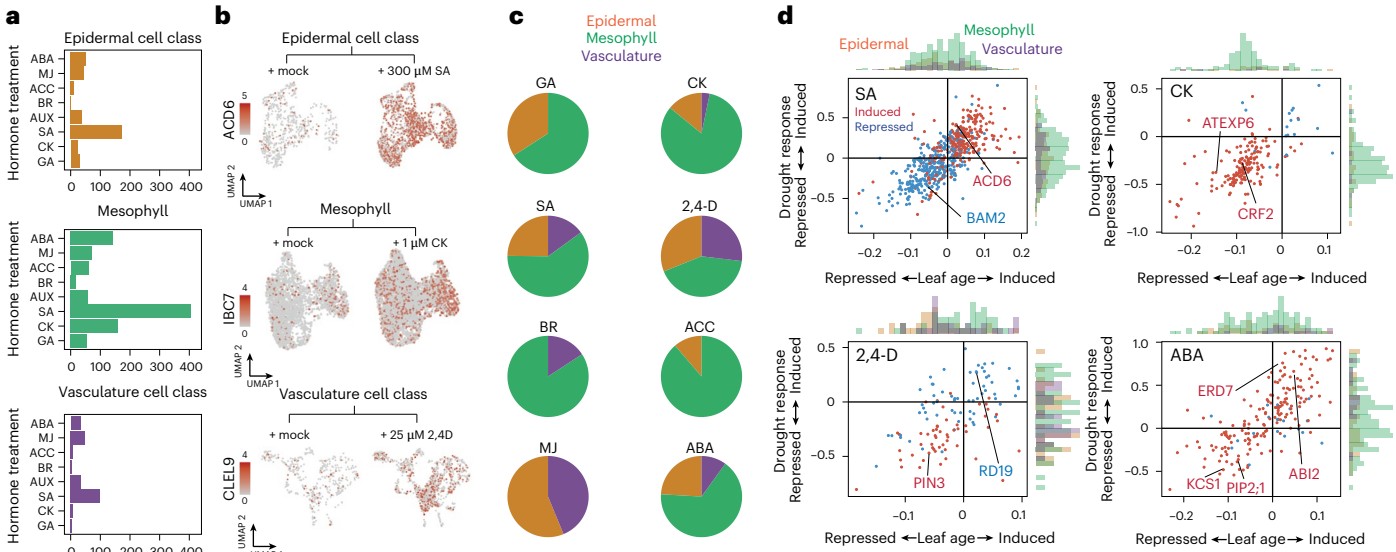

**Fig. 3 | Hormone signals are associated with drought-induced changes to leaf-ageing transcription dynamics. a,** Number of significantly differentially expressed (DE) genes in the mesophyll, the epidermal cell class (combining epidermis, guard and trichome cell types) or vasculature cell class (combining phloem, xylem, bundle sheath, hydathode and myrosin idioblast cell types) in response to treatment with ABA, MJ, ACC, BR, 2,4-D, SA, CK as *trans*-zeatin and GA. **b,** Single-nuclei expression responses to exogenous hormone treatment of *ACD6, INDUCED BY CYTOKININ 7 (IBC7)* and *CLE-LIKE 9 (CLEL9)* in epidermal, mesophyll and vasculature cell-type clusters, respectively. **c,** Pie charts show the relative distribution of hormone-responsive genes across epidermal, mesophyll and vasculature cell types for each hormone treatment. Only genes associated with leaf ageing and perturbed by drought are included. **d,** Induction (red) or repression (blue) of genes responsive to one of four hormone treatments, plotted against their corresponding induction or repression in response to leaf ageing or drought stress (axis units are coefficients of linear model). Histograms next to each axis show the cell-type class in which the hormone-responsive gene was detected.

leaf slows and senescence is activated prematurely[19]. This suggests that the transcriptional programmes underlying leaf ageing are responsive to environmental cues such as drought, yet this relationship has not been systematically studied. To investigate how drought alters cell-type-specific gene expression during leaf ageing, we repeated our leaf time-course experiment (Fig. 1a and Extended Data Fig. 1), however this time subjecting *Arabidopsis* rosettes to drought by withholding water for 9 days. This treatment reduced the water content (WC) within the vermiculite growth substrate from 100% to 21% and, as expected, led to smaller leaf sizes and reduced shoot biomass (Fig. 2a,b, Extended Data Fig. 5 and Supplementary Table 4). From this time course, we collected 579 drought-stressed leaves of varying ages and developmental stages (Extended Data Fig. 1). To capture drought-induced changes in cell-type-specific gene expression within these leaves, we again used the technique sci-RNA-seq3. By barcoding individual leaves during the reverse transcription step, we maintained the ability to resolve transcriptomes both by cell type and by leaf of origin across the drought time course. This approach yielded transcriptional profiles of 173,731 nuclei.

Across cell types, we found that drought stress induced the expression of genes upregulated during leaf ageing (Fig. 2c, Extended Data Figs. 4 and 6 and Supplementary Table 5). In the epidermis cell type, for example, examining gene expression under drought conditions revealed a subset of genes that displayed more advanced expression patterns compared with leaves of the same age under non-stress conditions (Fig. 2d and Supplementary Table 5; $P_{adj} < 0.01$, linear model). This effect appeared dose-dependent, with increasing stress severity (that is, decreasing water availability within the pot used for the plant) amplifying the shift in ageing-associated expression. The advancement of expression is exemplified by *RD29A*, which we found induced during leaf ageing (Fig. 1g) and advanced by drought stress (Fig. 2e). Similarly, the Calvin–Benson cycle gene *FRUCTOSE 1,6 BIPHOSPHATE ALDOLASE 2 (FBA2)*[20] was repressed by leaf ageing and showed earlier repression under drought (Fig. 2e). Such expression trends extended to other

genes involved in leaf development, the cell cycle and photosynthesis (Extended Data Fig. 6). Notably, drought stress led to epidermal nuclei from younger leaves to cluster more closely with those from older leaves (Fig. 2f and Extended Data Fig. 5). This suggests that drought stress prompts younger leaves to adopt expression profiles resembling those of older ones.

We observed similar trends in other cell types, most notably in the mesophyll. Here too, we found a subset of genes that were upregulated during leaf ageing advance their expression under drought stress (Fig. 2g). This included regulators of leaf size, such as *BRASSINOSTEROID INSENSITIVE 1 (BRI1)*[21] and *DA1*[22], which were downregulated or upregulated earlier within the mesophyll cell type under drought, respectively (Fig. 2h). Consistent with expression patterns we observed in the epidermis, drought stress also caused mesophyll nuclei from younger leaves to cluster more closely with those from older leaves (Fig. 2i). Together these findings indicate that a subset of genes involved in leaf ageing advance their expression prematurely when encountering stress.

## Drought stress impacts hormone-signalling responses to promote leaf ageing

Given the induction of ageing-associated gene-expression patterns under drought, we asked whether these responses agreed with known signals of leaf development. We hypothesized that hormone signalling, which plays a role in coordinating leaf development, may inform how drought stress alters ageing-associated expression patterns. To test this, we treated whole rosettes with eight different phytohormones for 2 h before sequencing transcriptomes of leaf tissue to single-nuclei resolution (Extended Data Fig. 1). Using this approach, we identified hundreds of differentially expressed genes responsive to specific hormone treatments in mesophyll, epidermal or vasculature cell-type classes (Fig. 3a,b and Extended Data Fig. 7). We then overlapped these responsive genes with those we identified as differentially expressed during leaf ageing or drought onset (Extended Data Fig. 8 and Supplementary Table 6).

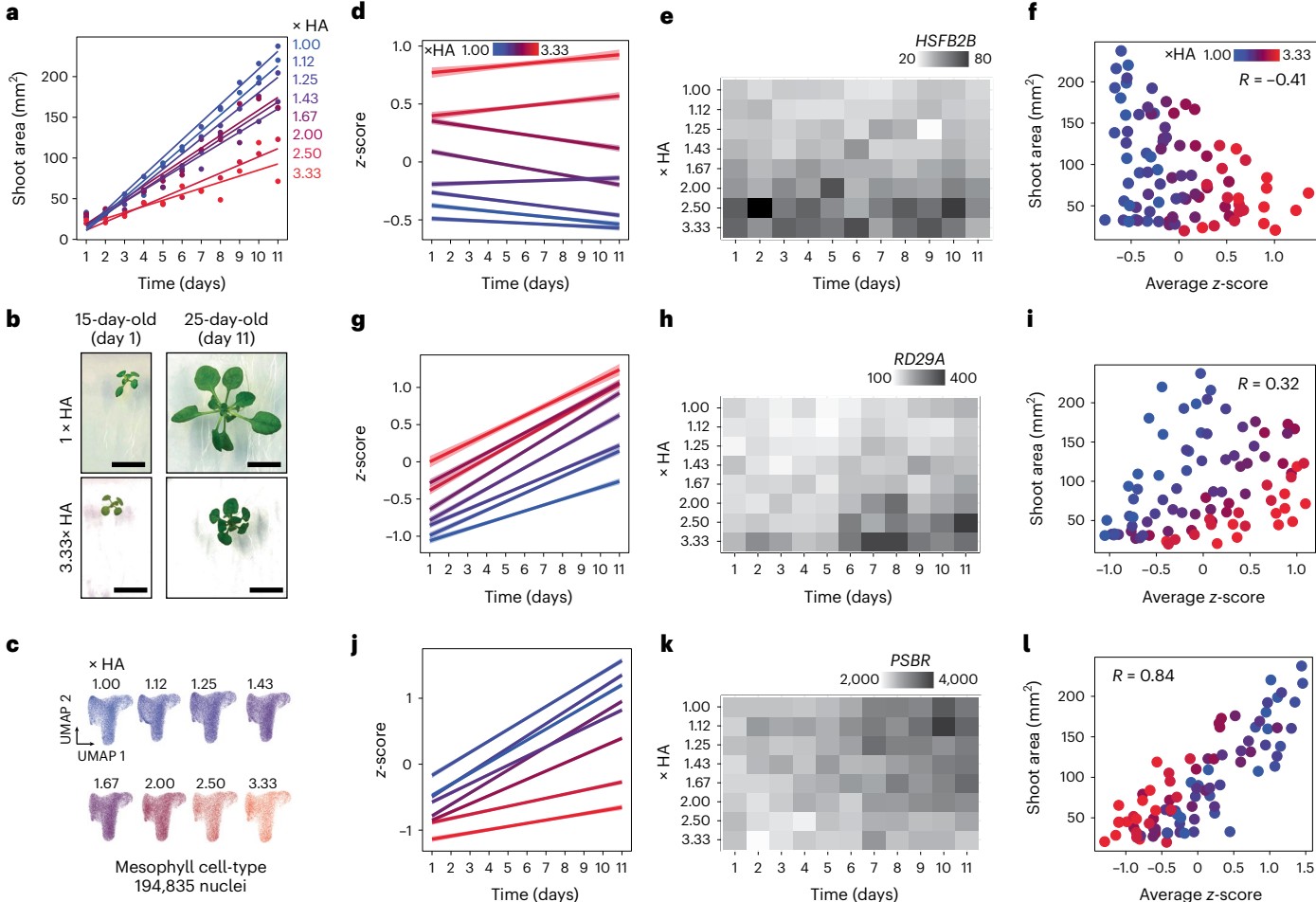

**Fig. 4 | Dose-responsive transcriptional changes in the mesophyll to HA stress are associated with shoot-size plasticity. a**, Shoot size of *Arabidopsis* seedlings grown for 11 days on eight different HA media doses. **b**, Images of seedlings on day 1 and day 11 grown under 1× (no stress) or 3.3× HA stress conditions (scale bar, 1 cm). **c**, A total of 595 individual shoots spanning the 11 days and eight HA doses were sequenced at single-nucleus resolution using sci-RNA-seq3. Clusters include 194,835 nuclei from the mesophyll cell type, separated by HA dose. **d**, Scaled expression of 1,964 genes whose expression in the mesophyll cell-type was induced in a dose-dependent manner with HA stress intensity (solid lines indicate linear model fits, shaded areas indicate 95% CIs). **e**, *HSFB2B* expression levels across the 88 conditions tested. **f**, Association between average scaled expression (mean *z*-score across 1,964 genes) and shoot area. **g**, Scaled expression of 511

genes whose expression in the mesophyll cell-type induced both over the 11-day time course and in a dose-dependent manner with HA-stress intensity (solid lines indicate linear model fit, shaded areas indicate 95% CIs). **h**, *RD29A* expression levels across the 88 conditions tested. **i**, Association between average scaled expression (mean *z*-score across 511 genes) and shoot area. **j**, Scaled expression of 858 genes whose expression in the mesophyll cell-type induced over the 11-day time course but repressed in a dose-dependent manner with HA-stress intensity (solid lines indicate linear model fit, shaded areas indicate 95% CIs). **k**, *PSBR* expression levels across the 88 samples tested. **l**, Association between average scaled expression (mean *z*-score across 858 genes) and shoot area.

Through these overlaps, we found that a subset of genes associated with leaf ageing and drought stress response were hormone responsive. Some hormone treatments, such as synthetic auxin (2,4-D), were associated with these expression patterns across mesophyll, epidermal and vascular cell types. By contrast, others, including CK and methyl jasmonate (MJ), elicited responses that were largely restricted to one or two cell-type classes (Fig. 3c).

Our results suggest that drought triggers several hormone-signalling pathways to induce cell-type-specific expression of genes associated with leaf ageing. For example, treatment with hormones known to be positively associated with leaf maturation largely induced expression of genes found responsive to both during leaf ageing and drought onset[19]. This included salicylic acid (SA), MJ, brassinosteroid (BR) and abscisic acid (ABA), with SA showing the strongest response (Extended Data Fig. 8). Consistent with the known role of SA in promoting senescence[23], SA-induced genes were significantly enriched among those upregulated during leaf ageing and further induced under drought (Fig. 3d). These genes were distributed across mesophyll, epidermal and

vasculature cell types and included the maturation regulator *ACCELERATED CELL DEATH 6* (*ACD6*)[24] in the epidermis and vasculature (Fig. 3b,d). Together, these findings suggest that drought triggers SA signalling to activate maturation programmes in a cell-type-specific manner. We note that ABA, MJ and BR enacted similar gene expression patterns, however, with different cell-type specificity (Extended Data Fig. 8).

Conversely, we also found evidence that drought stress suppressed hormone signals known to sustain leaf growth. Treatment with hormones known to be negatively associated with leaf maturation largely repressed expression of genes downregulated by leaf ageing and drought onset[19]. This included CK, gibberellin (GA) and 2,4-D (Extended Data Fig. 8). In line with the role of CK in promoting leaf expansion[13,25], CK-induced genes were significantly enriched among those downregulated both during leaf ageing and in response to drought (Fig. 3d). This repression was strongly biased towards the mesophyll, where 60% of affected genes (82 total) were related to protein translation. This indicates that drought suppresses CK signalling to limit cell expansion and promote ageing.

## Mesophyll gene expression in growing shoots scales with stress intensity

Among rosette leaves, we found evidence that drought advances the expression of genes associated with leaf ageing. Moreover, it appeared that this advancement may occur in proportion to the availability of water (Fig. 2d,g). Such dose-dependent transcriptional patterns may contribute to the stress-avoidance response, where shoot size declines proportionally as drought severity increases. Thus, we aimed to more precisely resolve the dynamics of this dose-dependent transcriptional response, focusing on the mesophyll, where these transcriptional changes appeared particularly robust (Fig. 2d,e,g,h). To this end, we exposed *Arabidopsis* seedlings to a range of controlled drought stress levels using the 'hard agar' (HA) system, which can afford more fine-scale adjustment of water potential compared with soil drying[26]. Seedlings were grown under eight HA stress levels for 11 days (Fig. 4a). As expected, shoot size declined in a dose-dependent manner as the dose of HA increased, reflecting the ability of *Arabidopsis* to initiate the stress-avoidance phenotype in proportion with stress severity (Fig. 4a). For example, during the time course, under unstressed (1× HA) conditions, seedling shoot area increased from 33 mm$^2$ in size to 220 mm$^2$, with this increase largely driven by leaves undergoing expansion (Fig. 4a). Under the most stressful conditions (3.33× HA), seedling shoot area increased only from 20 mm$^2$ to 71 mm$^2$ in size (Fig. 4b). To capture how mesophyll gene expression responded across these stress levels, we again used sci-RNA-seq3. This method enabled us to uniquely barcode and sequence nuclei from each of the 88 experimental conditions (eight HA doses, 11 days, totalling 595 individual shoots) and thus the ability to map each sequenced nucleus to its specific HA dose and time point. In total, we profiled 456,008 shoot nuclei, including 194,835 mesophyll nuclei (Fig. 4c and Extended Data Fig. 1).

Within the mesophyll cell type, we detected three distinct expression trends in seedlings as they grew under increasing levels of HA stress. First, we identified 1,964 genes that were induced in a dose-dependent manner in response to HA stress, but which remained unchanged over the 11-day period (Fig. 4d) ($P_{adj} < 0.01$, linear model). These genes were significantly enriched in drought-associated GO functions such as 'response to water deprivation' ($P_{adj} = 7.64 \times 10^{-4}$) and included gene *HEAT STRESS TRANSCRIPTION FACTOR B-2b* (*HSFB2B*) (Fig. 4e). By contrast to genes that did not change over time, we also identified a second class of 511 HA dose-responsive genes whose expression also increased progressively as shoots grew over the 11 days (Fig. 4g). Genes within this class were also enriched in drought-associated functions (for example 'response to osmotic stress' GO term ($P_{adj} = 1.6 \times 10^{-7}$)). This expression trend suggests that HA stress can proportionally activate genes induced during shoot development. This pattern reflects similar trends we observed among rosette leaves. For instance, the senescence-associated gene *RD29A* showed a time-dependent increase in gene expression under HA stress (Fig. 4h), mirroring its induced expression in response to leaf ageing and drought stress in rosette leaves (Fig. 2e). A third and distinct class of 858 genes were induced during seedling development but repressed in proportion to HA stress severity (Fig. 4j). Genes within this class were enriched for 'photosynthesis' and related GO terms ($P_{adj} = 1.8 \times 10^{-40}$), among others. For instance, expression of the photosynthetic apparatus gene *PHOTOSYSTEM II SUBUNIT R* (*PSBR*) was induced as seedlings grew over time, but declined proportionally with increasing stress severity (Fig. 4k).

Crucially, when we correlated the expression patterns of each of these three classes of gene expression patterns with shoot size, we found that the expression of 858 genes that were induced during seedling development but repressed by HA stress (the third class), were most strongly associated with changes in shoot size (Fig. 4l) (Pearson $R = 0.84$, $P < 2.2 \times 10^{-16}$). This suggests that these 858 genes may, in part, contribute to the stress-avoidance response phenotype. We note that similar gene expression trends were observed across other cell types (Supplementary Table 7).

## Mesophyll-specific FRO6 modulates stress-responsive shoot growth

Having identified gene expression trends linked to the stress-avoidance phenotype, we asked whether any individual gene plays a causal role. We focused on the 858 mesophyll-expressed genes whose expression most strongly correlated with seedling shoot size under HA stress (Fig. 4l). To identify a candidate, we followed two criteria. First, we correlated the expression of each gene with seedling size in response to HA stress (Fig. 5a). Second, to independently confirm that a gene was drought responsive, we assessed whether they were differentially expressed in rosette leaves subjected to drying in pots (Fig. 5a). From this analysis, we identified FRO6 (Fig. 5a,b). FRO6 is a membrane-bound ferric chelate reductase that converts iron(III) to iron(II)[27] and is hypothesized to transport iron(II) to chloroplasts for energy production[28]. FRO6 expression was strongest in the mesophyll cell type and, according to our atlas, FRO6 was induced during leaf ageing, but repressed by drought (Fig. 5b,e). When we examined subcellular localization of FRO6 using a fluorescent reporter line (*FRO6p::FRO6–GFP*), it appeared to be localized to the endoplasmic reticulum (Fig. 5c)[29]. We confirmed this localization pattern with an endoplasmic reticulum reporter line (Extended Data Fig. 9)[30].

To counter drought-induced transcriptional repression of *FRO6* within the mesophyll cell type, we leveraged our transcriptional atlas to identify promoters with high expression in the mesophyll. Among these, the *TPR-DOMAIN SUPPRESOR OF STIMPY* (*TSS*, also called *REDUCED CHLOROPLAST COVERAGE 2*)[31] stood out for its mesophyll-enriched expression that exceeded *FRO6* amounts by approximately twofold (Fig. 5e and Extended Data Fig. 9). We note *TSS* itself was neither significantly repressed by drought nor by HA stress ($P_{adj} > 0.05$, linear model), nor was *TSSp* active within root tissue (Extended Data Fig. 9), however it did show lower expression in other leaf cell types within our atlas (Fig. 5e). We overexpressed *FRO6* in a mesophyll-targeted manner using the 1,643 base pairs (bp) upstream region of *TSS*.

Compared with wild-type plants (Col-0), we observed that two independent transgenic *TSSp::FRO6–GFP* lines (allele 1 and allele 2) grown on soil had significantly increased shoot dry weight under drought conditions (Fig. 5f, $P < 0.05$, Welch's one-sided *t*-test). Similarly, when we grew these transgenic lines on vermiculite, we observed increased shoot biomass and leaf area under drought conditions (Fig. 5d,g,h), with *TSSp::FRO6–GFP* allele 1 significantly different ($P < 0.05$) and allele 2 marginally significant ($0.05 < P < 0.1$). This trend persisted when we examined the shoot size of transgenic lines grown on HA stress (Fig. 5i). Notably, the shoot biomass and leaf area of the transgenic lines did not increase significantly under non-stress conditions ($P > 0.1$, Fig. 5f–i) These findings support the hypothesis that mesophyll-specific repression of *FRO6* mediates shoot growth plasticity under drought, and demonstrate that its overexpression in this cell type is sufficient to partially counteract the stress-avoidance response.

## Discussion

Upon encountering drought, *Arabidopsis* advances transcriptional responses related to leaf ageing to limit shoot growth. This finding helps unify earlier reports detailing the impact of drought on leaf development. Elegant studies have shown that drought modifies gene regulation to restrict proliferative and expansive phases, arrest the cell cycle and hasten the transition to endoreduplication, along with activating senescence-associated genes to expedite leaf death[1–3,8,32]. Here we find evidence that these distinct leaf developmental responses to drought can be understood as early induction of leaf-ageing responses, leading to induced leaf ageing. These results agree with discoveries made in epigenetics. For example, the advancement of leaf biological age

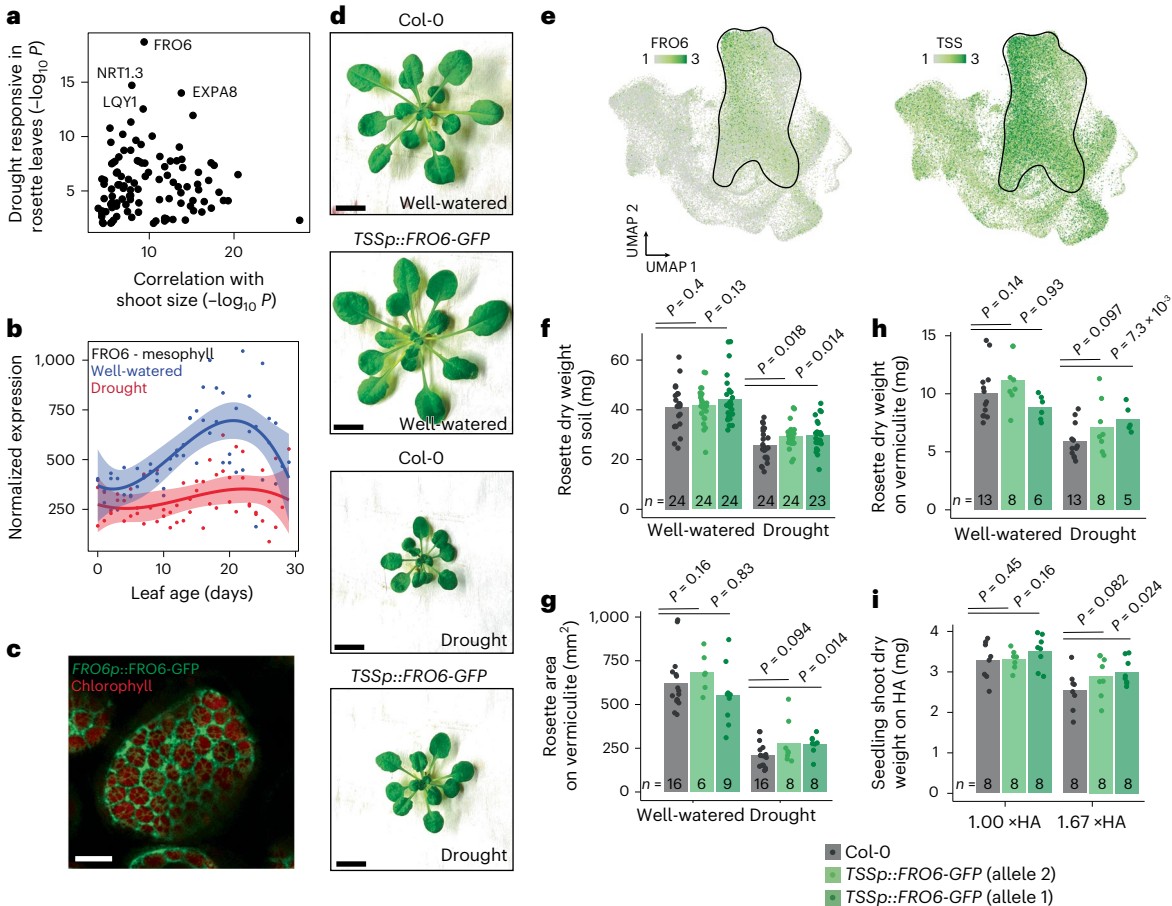

**Fig. 5 | FRO6 within the mesophyll modulates shoot growth under drought stress. a**, Scatter plot of mesophyll-expressed genes, with each gene positioned according to two independent statistical measures—the correlation between gene expression and shoot size under HA stress, and the responsiveness of the gene to drought in pot-grown rosettes. **b**, Mesophyll FRO6 transcriptional abundance under well-watered (100% WC, blue) and drought conditions (22–48% WC, red) across leaf age. Curves fit using a quadratic model with 99% CI indicated by shaded area. **c**, Confocal microscopy showing *FRO6::FRO6–GFP* and chlorophyll localization within a mesophyll cell (10-μm size indicated). **d**, Representative images of Col-0 and *TSSp::FRO6–GFP* rosettes grown on vermiculite under either well-watered or drought conditions (1 cm size

indicated). **e**, Single-nucleus expression localization of FRO6 and TSS across leaf nuclei. Black outline denotes the mesophyll cluster. **f**, Whole-rosette dry weight of Col-0 and two independent *TSSp::FRO6–GFP* transgenic alleles grown on soil under well-watered or drought conditions (bars indicate sample mean, points represent individual rosettes, Welch's one-sided *t*-test *P* indicated). **g,h**, Whole-rosette dry weight (g) or rosette area (**h**) of Col-0 and two *TSSp::FRO6–GFP* alleles grown on vermiculite under well-watered or drought stress (bars indicate sample mean, points represent individual rosettes, Welch's one-sided *t*-test *P* indicated). **i**, Shoot dry weight of Col-0 and *TSSp::FRO6–GFP* seedlings grown on unstressed (1× HA) or stressed (1.67× HA) conditions (bars indicate sample mean, points represent individual seedlings, Welch's one-sided *t*-test *P* indicated).

ahead of calendar age was recently described at the epigenetic level[33]. Similarly, evidence points to the stress-avoidance response being in part regulated epigenetically[34].

A defining feature of the stress-avoidance response is its ability to limit shoot growth in proportion to the intensity of stress. The dose-dependent transcriptional responses to stress we describe here are associated with this plasticity. Examples of dose-responsive transcriptional changes and their relationship to organ size are observed elsewhere, such as in response to nutrient availability in *Arabidopsis* roots[35] and water stress responses in *Arabidopsis* seedlings and rice shoots[36,37]. We find that such dose-responsive gene expression is, in part, dependent on hormone signalling. Specifically, we find that genes responsive to CK and SA treatment—hormones whose concentrations are known to prolong or restrict leaf lifespan, respectively[23,25]—are well represented among the leaf-ageing-associated genes that change in response to drought stress. Moreover, by targeted perturbation of *FRO6* expression in mesophyll cells, we demonstrate that the stress-avoidance response can be altered. This highlights the mesophyll as a site of stress signalling during organ development—a role not commonly attributed to this cell type. Further study may reveal the

mechanism by which FRO6 mediates shoot size under stress. Similarly, given the activity of FRO6 in the mesophyll, future work might explore the role that chloroplast signalling plays in stress perception[38].

Our findings have broader implications for the development of drought-resilient crops. For instance, substantial efforts have focused on engineering crop varieties that can withstand drought stress while maintaining optimal leaf growth under non-stress conditions[39]. This has proven to be challenging, as candidate genes identified through drought response assays often improve drought resilience by limiting of overall plant stature[40,41]. Here we find that cell-type-specific overexpression of *FRO6*, a gene repressed by drought but induced during leaf ageing, weakened the stress-avoidance response in *Arabidopsis*. Our transcriptional atlas of leaf development may facilitate similar engineering strategies in the future.

## Methods

### *Arabidopsis* growth conditions for physiological measurements and single-nuclei transcriptome sequencing

*Arabidopsis* Col-0 seeds were surface-sterilized and stratified for 2 days, and then grown for 17 days on LS medium (Cassion). Plates were

supplemented with 1% sucrose and incubated under short-day conditions (8 h light) with light intensity set at 150 µmoles at 22 °C. Short-day conditions were chosen to prevent flowering, thereby ensuring that our study captured the drought-stress-avoidance response rather than the drought-stress-escape response[4]. After this period, plants were transferred to vermiculite and grown on 0.75× LS media without sucrose for 12 days, maintaining constant saturation. On the 12th day, the first leaf samples were taken for sci-RNA-seq3 sequencing and collected 4.5 h after subjective dawn. To introduce drought stress, excess water was removed until all pots reached 100% field capacity (FC)—that is, the condition where the vermiculite could hold the maximum amount of water without dripping. We note that in the context of this manuscript, 100% FC is synonymous with 100% WC. Subsequent leaf sampling was conducted daily for 8 days at the same time of day and the degree of evaporation was measured by weighing each pot. Three rosettes were sourced from drought-stressed plants and three from well-watered control groups per time point. Two biological replicates were performed (that is, two distinct drought-stress time-course experiments performed on two separate occasions). At the time of sampling, whole rosettes were flash-frozen and then each leaf was excised and placed into a 96-well plate. Leaf measurements were conducted using Plant Growth Tracker software (https://github.com/jiayinghsu/plant-growth-tracker/tree/main) (Supplementary Table 4). We applied an analysis of covariance (ANCOVA) model (designating drought stress and time as qualitative and quantitative variables, respectively) to assess statistical differences between rosette shoot size. We note that *TSSp::FRO6–GFP* genotypes were grown under the same vermiculite growth conditions as described above where two plants per genotype (one wild-type, one transgenic) were grown in the same pot. For soil-grown experiments, Col-0 and two independent *TSSp:FRO6* lines were stratified for 4 days and then spotted onto presoaked soil (day 0). Seeds germinated by day 2. Plants were grown in long-day conditions (16 h light/8 h dark) at 22 °C under approximately 100 µmol m$^{-2}$ s$^{-1}$ of white light. Two plants per genotype were grown in the same pot (total six plants per pot) across 24 pots. By day 12, soil moisture had begun to decline; at this stage, well-watered pots were rewatered (12 pots), while drought-treated pots remained unwatered (12 pots). Drought pots received their first watering on day 14 (50 ml per pot), at which point soil was dry but plants showed no visible drought symptoms. Two additions of 50 ml of water occurred on day 17 and day 20, each separated by a 3-day dry period. After a final 3-day dry period, plants were harvested on day 23. Whole rosettes were excised and dried at 55 °C for dry-weight analysis.

### *Arabidopsis* growth conditions for exogenous hormone treatment

*Arabidopsis* seedlings were sterilized and stratified as described above before being grown on vertical plates for 17 days under short-day conditions on 1× LS media supplemented with 1% sucrose. After this time, seedlings were transferred to vermiculite (0.75 LS media, no sucrose). Plants were grown at complete FC saturation for 18 days. On the 18th day, 2 h after subjective dawn, both roots (through replacing growth media) and shoots (through foliar spray) were treated with one of the following hormone treatments: 10 µM (±)-ABA, 25 µM 2,4-D, 50 µM GA$_3$, 300 µM SA, 1 µM *trans*-zeatin (CK), 10 nM BR, 10 µM MJ, 50 µM 1-aminocyclopropane-1-carboxylic acid (ACC). Each hormone solution contained 0.01% DMSO and 0.1% ethanol. A mock treatment control was also included. Plants were treated for ~3 h before the rosettes were flash-frozen in liquid nitrogen. About 15 plants were collected per treatment.

### *Arabidopsis* growth conditions for HA treatment

*Arabidopsis* Col-0 seeds were sterilized and stratified as described above before being grown on vertical plates supplemented with 1× LS media, 1% sucrose and 2% agar with a light period of 8 h (150 µmoles) at 22 °C. We designated 2% agar and 1× LS media as the '1× HA' dose. Thus, a '2× HA' dose consisted of 4% agar and 2× LS media. Additional description and

validation of HA stress are described in ref. 26. Sampling of individual seedlings for sci-RNA-seq3 began on the 15th day, 4.5 h after subjective dawn and continued for a total of 11 days. Plants were imaged on plates for shoot area measurements before excising the roots and flash-freezing each individual shoot in a 96-well plate (*n* = 7 individual seedlings per time point, per HA dose). Images were processed using Plant Growth Tracker software (Supplementary Table 4). We note that for *TSSp*::FRO6 shoot area measurements, the same HA growth protocol was followed as described above; however, images were taken 29 days after sowing (*n* = 6 individual seedlings across two plates, per treatment, per genotype).

### Nuclei extraction and single-nuclei RNA sequencing (sci-RNA-seq3)

The sci-RNA-seq3 was performed as described in ref. 11 with the following notable exceptions. Each frozen leaf or seedling sample was bead bashed (Qiagen) in a 96-well plate format. The resulting frozen homogenate was resuspended in resuspension buffer (10 mM Tris-HCl pH 7.4, 10 mM NaCl, 3 mM MgCl$_2$, 1% PBS, 0.5% DEPC). Tissue samples were then passed through a 96-well, 30-µm filter. Washed nuclei were concentrated and nuclear RNA reverse-transcribed with a well-specific primer. Subsequent ligation, tagmentation and PCR steps of sci-RNA-seq3 were followed as described in ref. 11. Libraries were sequenced on the Illumina Novaseq 6000 with 150-bp paired-end chemistry. The resulting reads were aligned to the *Arabidopsis* TAIR10 genome with Araport11 annotation[42]. The number of nuclei sequenced per sample and unique molecular identifiers (UMIs) per nucleus are reported in Supplementary Table 1.

### Nuclei extraction and single-nuclei RNA sequencing (10× RNA-seq)

Nuclei isolation was performed upon whole rosettes. Frozen tissue was crushed using a mortar and pestle and nuclei released from the homogenate using a resuspension buffer (10 mM Tris-HCl pH 7.4, 10 mM NaCl, 3 mM MgCl$_2$, 1% PBS, 1% superase RNAse inhibitor). The resulting homogenate was filtered using a 30-µm filter. To enrich for nuclei, an Optiprep (Sigma) gradient was used. Enriched nuclei were then purified using fluorescent activated cell sorting. Purified nuclei were loaded directly onto the 10× machine 10X-Gene Expression v.3.0 chemistry and sequenced on the Illumina Novaseq 6000 with 150-bp paired-end chemistry. Libraries were sequenced and aligned to the *Arabidopsis* TAIR10 genome with Araport11 annotation[42]. Chloroplast and mitochondrial reads were removed. The number of nuclei per sample and UMI per nucleus are reported in Supplementary Table 1.

### Nuclei clustering

Transcriptional atlases from each experiment were assembled using Seurat[43]. Nuclei were first subsetted using a minimum UMI threshold of 450 reads. Then, nuclei from various experiments (individual rosette leaves, individual seedlings and hormone treatments) were combined into a single atlas. The integrated dataset was subjected to clustering, using the top 3,000 variable features that were shared across all datasets. Subsequent uniform manifold approximation and projections (UMAPs) were constructed using the first 30 principal components. Very small clusters were considered artefacts and removed. Cell types were subsequently annotated using marker genes listed in Supplementary Table 2. The exception was the bundle-sheath cell-type cluster. Here we used the bundle-sheath cell-type-specific markers described in ref. 44 (with an enrichment score >6) to identify the bundle-sheath cell type within our atlas. This was done by assessing how many of these published bundle-sheath marker genes were the same as the cluster-specific genes present in our atlas (with an enrichment score >3). We note that UMAP projections of cell types under well-watered conditions (100% WC) or drought conditions (<39% WC) were performed using the same approach as described above with the notable exception that we downsampled nuclei to ensure equal numbers of nuclei were present across

well-watered and drought samples of nuclei within each leaf stage and used genes found differentially expressed during leaf development or responsive to drought stress as variable features.

### Detecting cell-type-specific leaf maturation gene expression patterns and their stress response

To detect genome-wide changes in gene expression during leaf maturation, we established two criteria—first, a gene needed to be differentially expressed across different leaf stages within a rosette. Second, a gene also needed to be differentially expressed over time as each leaf matured. We used a linear model to test these criteria. Initially, we examined gene expression patterns in whole leaf organs using a multivariate linear model:

$$\text{gene expression}_a = \text{leaf stage} + \text{time} + \text{drought} + c$$

Where $a$ indicates a gene's normalized expression counts, and $c$ indicates the intercept for the linear model. This analysis, which included data from both well-watered and drought-stressed conditions, indicated that the factors 'leaf stage' and 'time' were colinear. Thus, we combined these two variables into a single 'leaf age' variable, defined by the following formula:

$$\text{leaf age} = 23 + \text{time} - (\text{leaf stage} \times 2) + c$$

Following this, we modelled gene expression profiles for each cell type using a simplified model:

$$\text{gene expression}_a = \text{leaf age} + \text{drought} + c$$

where both time and leaf stage were quantitative variables (Extended Data Fig. 4). This multivariate linear model was implemented in DESeq2[45] on the pseudobulked gene expression profile of each cell type, where genes were first quantile normalized. We classified a gene as involved in maturation if the leaf age coefficient was significant ($P_{adj} < 0.01$, leaf age coefficient >0.04) and as drought responsive if the drought coefficient was significant ($P_{adj} < 0.01$).

### Detecting hormone-responsive genes among cell-type classes

Genes found differentially expressed in response to exogenous hormone treatment were detected by comparing each hormone treatment to the mock control, using the FindMarkers() command in Seruat[43] ($P_{adj} < 0.05$; differential expression was assessed using a likelihood ratio test, comparing a full model that included group identity to a reduced model lacking the group term). Because less abundant cell types did not have enough nuclei to perform robust statistical testing, we combined vasculature cell types into a 'cell class' (phloem, xylem, bundle sheath, hydathode and myrosin idoblast cell types). We used the same approach to combine epidermal cell types (epidermal, guard and trichome cell types). In each cell class, we found some genes that were differentially expressed in response to several hormone treatments. To ensure that we only analysed genes responsive to specific hormones, for each cell class, we removed genes that were differentially expressed in response to three or more hormones. We intersected our final set of hormone-responsive genes with those that were found to be differentially expressed during leaf maturation or in response to drought within our individual leaf rosette experiment (linear model, $P_{adj} < 0.01$). We only retained genes whose intersect was significant for further analysis (Fisher exact test $P < 0.05$, using a background of all expressed genes). To ensure greater stringency of statistical testing, intersections were performed with directionality. GO terms were called using agriGO software, with the whole genome as background.

### Detecting dose-responsive, cell-type-specific transcriptional responses to HA stress

We used a multivariate linear model to identify genes that were both differentially expressed during the 11 days of *Arabidopsis* seedling shoot growth and dose-responsive to the level of HA stress. This was achieved by first pseudobulking the expression profile of each cell type across each of the 88 conditions assayed (11 time points, eight HA doses) and removing low-count reads. Then, the statistical model was implemented in DESeq2[45] using quantile-normalized reads, with time and HA dose considered as quantitative variables. A gene was identified as significantly differentially expressed in both factors at an adjusted $P$ value threshold of 0.01. This modelling approach was implemented for six of the major leaf cell types identified and the resulting lists of significant genes are presented in Supplementary Table 7. We binned genes as either HA stress induced or HA stress repressed by relying on whether the coefficient of the HA factor was positive or negative, respectively.

### Plasmid construction

A 4,198-bp genomic fragment from the *FRO6* locus, which contains the *FRO6* coding region, 886 bp upstream of ATG and 211 bp of 3′ UTR, was used to generate the *FRO6p*::FRO6:GFP transgenic line. GFP was translationally fused to the C terminus of *FRO6* before the stop codon. For the *TSSp*::FRO6:GFP transgenic line, a 1,643-bp fragment upstream of the TSS start codon was used to replace the FRO6 promoter. The same TSSp fragment was used in generating the *TSSp*::GUS transcriptional fusion. All transgenes were cloned into the binary vector pMX202 and stably transformed into the *Arabidopsis* Col-0 background. The next generation of the same Col-0 background seedstock was grown alongside *TSSp*::FRO6:GFP transgenic lines and used for comparison. To generate the 2x*35S*::mCherry-HDEL plasmid, gateway cloning (Invitrogen) was used. The mCherry-HDEL coding sequence was PCR amplified from mCherry template using primers 5′-GGGGACAA-GTTTGTACAAAAAAGCAGGCTccATGGTGAGCAAGGGCGAGGAGGAT-3′ and 5′-GGGGACCACTTTGTACAAGAAAGCTGGGTcTTAAAGCTCAT-CATGCTTGTACAGCTCGTCCATGCCGC-3′. This was recombined into pDONR 221 before recombining with the destination binary vector pK7m34GW in addition to pDONR P4P1R carrying 2x*35S* sequences and pDONR-P2RP3 carrying a random 25-bp sequence[46].

### Confocal microscopy

Cotyledons of 6-day-old seedlings carrying the *FRO6*::GFP fusions were imaged with a Leica Stellaris 8 confocal microscope, using 473-nm laser excitation. GFP fluorescence signals were collected between 480 nm and 580 nm, with a lifetime range of 5.1–9 ns to exclude the chlorophyll autofluorescence signal. The chlorophyll autofluorescence was collected between 600 nm and 670 nm. To assess HDEL localization, 5-day-old cotyledons of the $F_1$ plants carrying both *TSSp*::FRO6:GFP and *35S*::mCherry:HDEL were imaged. The FRO6–GFP fusion was excited at 489 nm and the signal was collected at 500–570 nm with a lifetime range of 5.1–9 ns. The mCherry-HDEL was excited at 587 nm and the signal was collected at 600–635 nm with a lifetime range of 5.1–9 ns. Chlorophyll autofluorescence was excited at 405 nm and emission signal was collected at 650–750 nm.

### GUS staining

GUS activity staining was carried out as described in ref. 47, using 10 mM potassium ferro and ferri cyanide. The GUS-stained seedlings were mounted in 30% glycerol and imaged using a Zeiss Axio Zoom. V16 stereomicroscope equipped with an Axiocam 305 colour camera.

### Reporting summary

Further information on research design is available in the Nature Portfolio Reporting Summary linked to this article.

## Data availability

Single-cell RNA-seq data are publicly available through GEO (GSE290214). The atlas can be observed and interrogated at https://neomorph.salk.edu/SCMDAP/LeafDevUnderDrought. Unprocessed microscopy images are available via Figshare at https://doi.org/10.6084/m9.figshare.31216066 (ref. 48).

## Code availability

The code for bioinformatic analyses presented in this manuscript is available on GitHub at https://github.com/joey1463/Leaf-by-Leaf-Drought-Atlas.git.

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

## Acknowledgements

J.S. is an Open Philanthropy awardee of Life Science Research Foundation, as well as recipient of the Pratt Industries American-Australian Association Scholarship. J.R.E. and J.C. are Investigators at the Howard Hughes Medical Institute.

## Author contributions

J.S., J.R.E., X.W. and J.C. designed the experimental plan. J.S. performed the nuclei sequencing experiments and genomic analyses. X.W. performed transgenics and microscopy. C.P. cloned the marker line for microscopy. J.S. and X.W. performed physiological measurements. J.X. and N.I.-E. developed and implemented the Shoot Growth Tracker software. T.J. assisted in bioinformatics. J.R.N. performed sequencing workflows. J.S. and J.R.E. wrote the paper.

## Competing interests

J.S. is a co-founder of Crop Diagnostix Inc., a company that provides sequencing services. The mission of Crop Diagnostix is unrelated to the research presented in this paper. The other authors declare no competing interests.

## Additional information

**Extended data** is available for this paper at https://doi.org/10.1038/s41477-026-02254-3.

**Correspondence and requests for materials** should be addressed to Joseph Swift or Joseph R. Ecker.

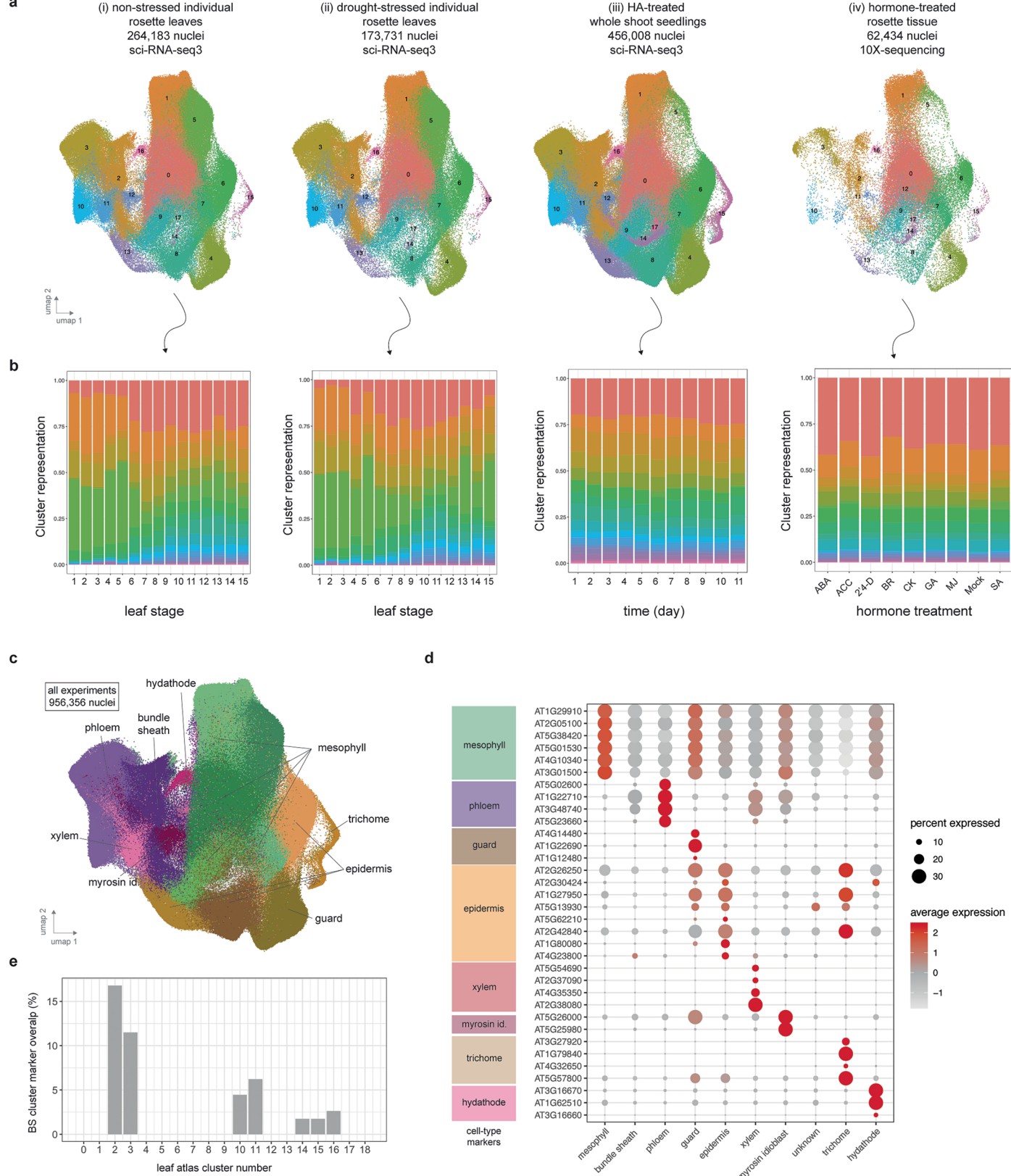

**Extended Data Fig. 1 | An Arabidopsis leaf transcriptional atlas. (a)** A total of 956,356 nuclei were sequenced across 4 different experiments: (i) non-stressed individual rosette leaves, (ii) drought-stressed individual rosette leaves, (iii) whole shoot seedlings grown on HA agar, and (iv) whole rosette tissue treated exogenously with hormones. All four experiments were integrated into a single atlas before identifying Arabidopsis leaf cell types. **(b)** Cluster representation for each leaf stage (rosette experiments), time of sampling (seedling experiment), or hormone treatment. **(c)** Cell-type annotation for each cluster within the atlas. **(d)** Cell types were annotated by relying on the expression of validated cell type marker genes (see Supplementary Table 2). **(e)** The bundle sheath cell type (cluster 2) was identified by overlapping the significant bundle sheath markers found in Kim et. al 44 with the cluster-specific marker genes from this study.

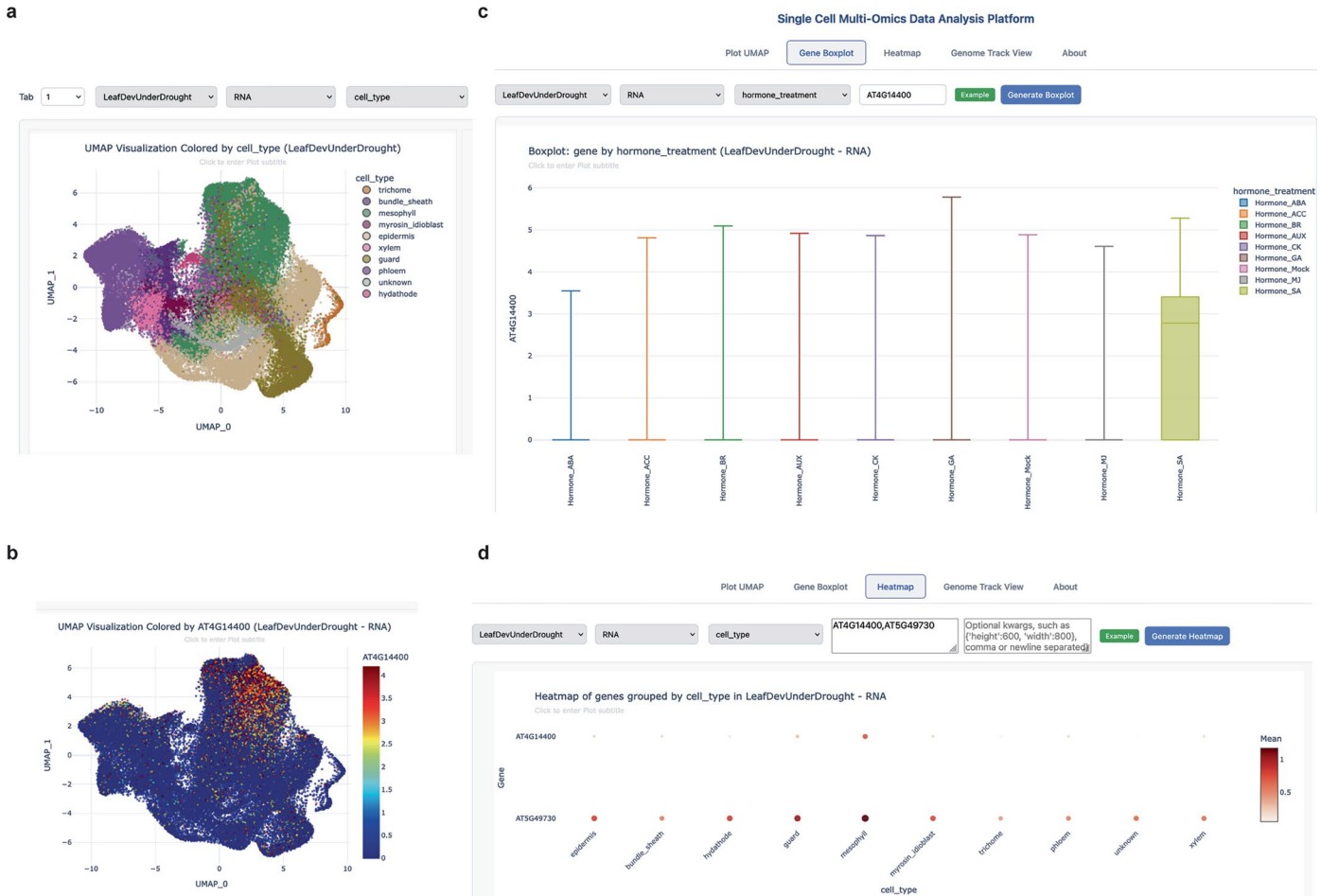

**Extended Data Fig. 2 | The Arabidopsis leaf atlas is available online. The transcriptomic atlas holding the different experiments presented in this study is available online.** This interface enables users to view how (**a**) nuclei cluster by cell type as well as (**b**) visualize individual gene expression (the gene ACD6 is illustrated). (**c**) Moreover, gene expression responses to different exogenous hormone treatments can be interrogated (**d**) and the cell type specific expression patterns between genes examined. Additional analyses can be initiated by the user. These data can be accessed at https://neomorph.salk.edu/SCMDAP/LeafDevUnderDrought.

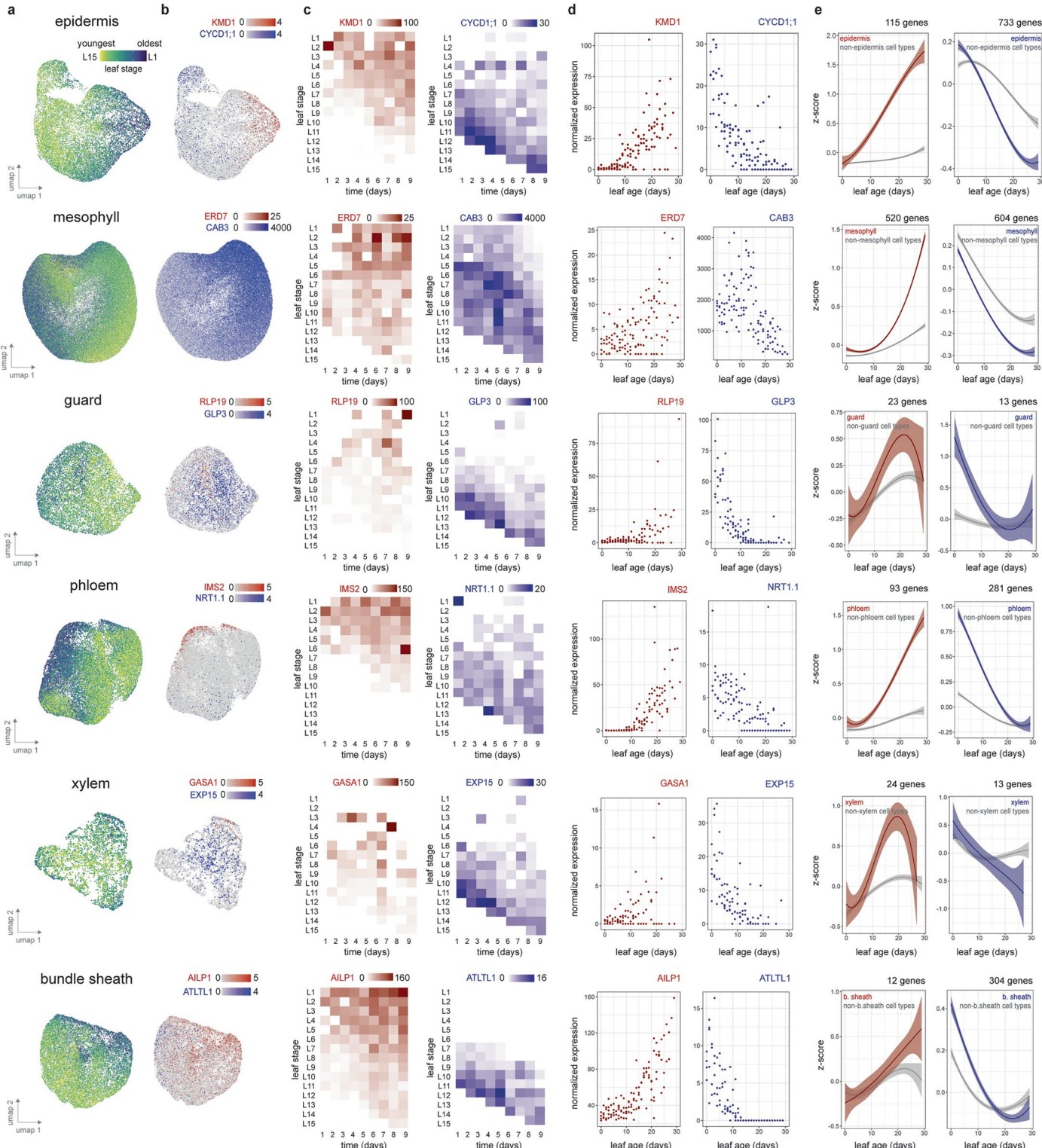

**Extended Data Fig. 3 | Cell-type specific transcriptional responses underlie Arabidopsis leaf maturation.** (**a**) Sub-clustering of 6 leaf cell types, with nuclei colored by the leaf maturation stage from which they originate. (**b**) Expression profile of selected genes that were either upregulated (red) or downregulated (blue) during leaf maturation within each cell type. (**c**) Cell-type specific pseudo-bulked expression profile of selected genes ordered by leaf stage and day of sampling. (**d**) Expression profile of selected genes as leaves aged and matured. (**e**) z-score normalized expression trend of cell-type specific genes that fit a linear model with an adj. p < 0.01, that were either significantly induced or repressed during leaf maturation (curve fit using quadratic model (solid line), 99% CI indicated (shaded area)).

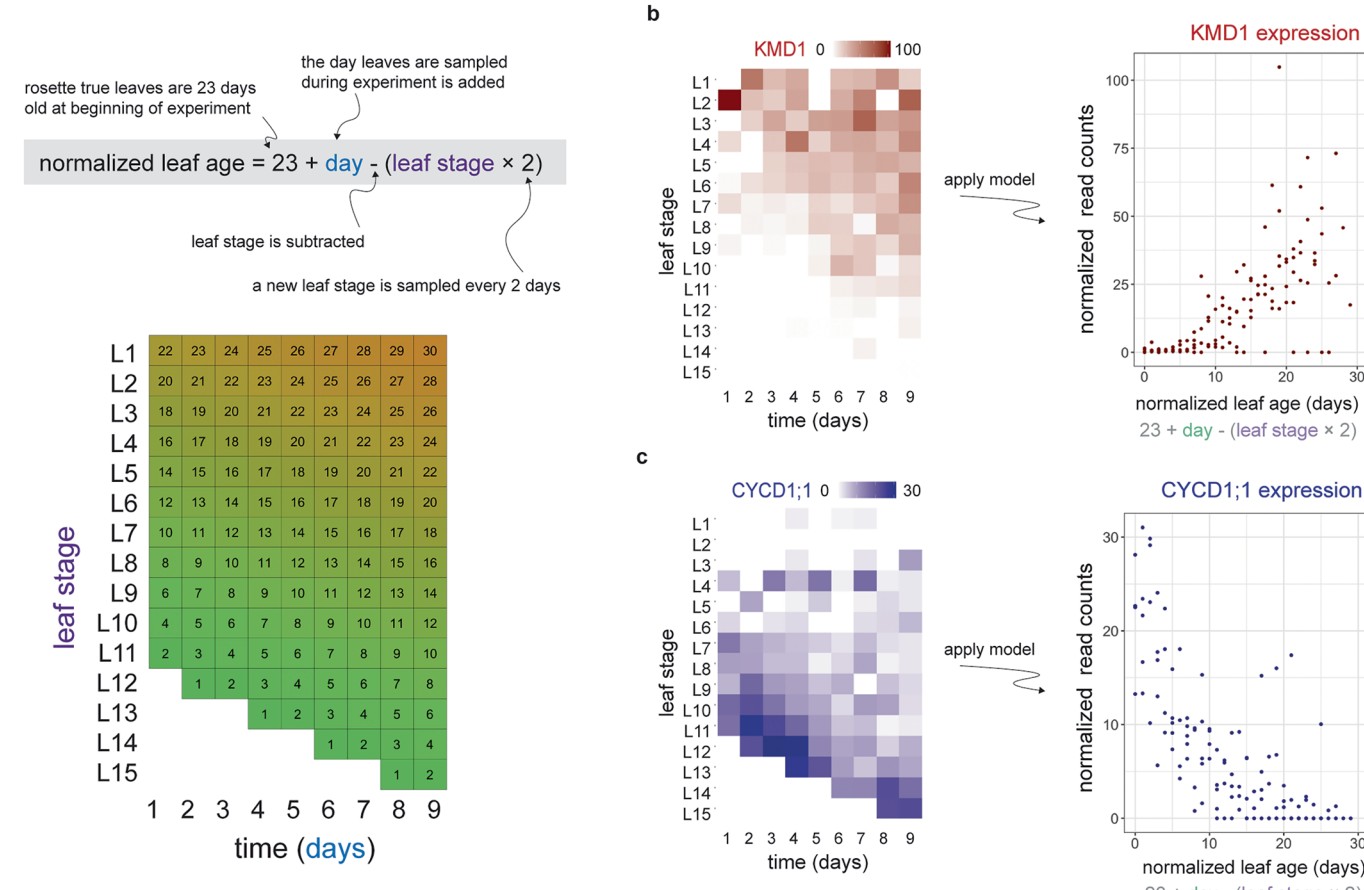

**Extended Data Fig. 4 | Computing a normalized leaf age value based on a leaf's developmental stage sampling time. (a)** Normalized leaf age was calculated by taking into account the age of the rosette leaves at the start of the experiment, the sampled time point, and the stage of the sampled leaf. The application of this model to each leaf sample is shown below, where each grid number represents a leaf's normalized age value. As an example, this model is applied to the expression patterns of two example genes (**b**) KMD1 and (**c**) CYCD1;1.

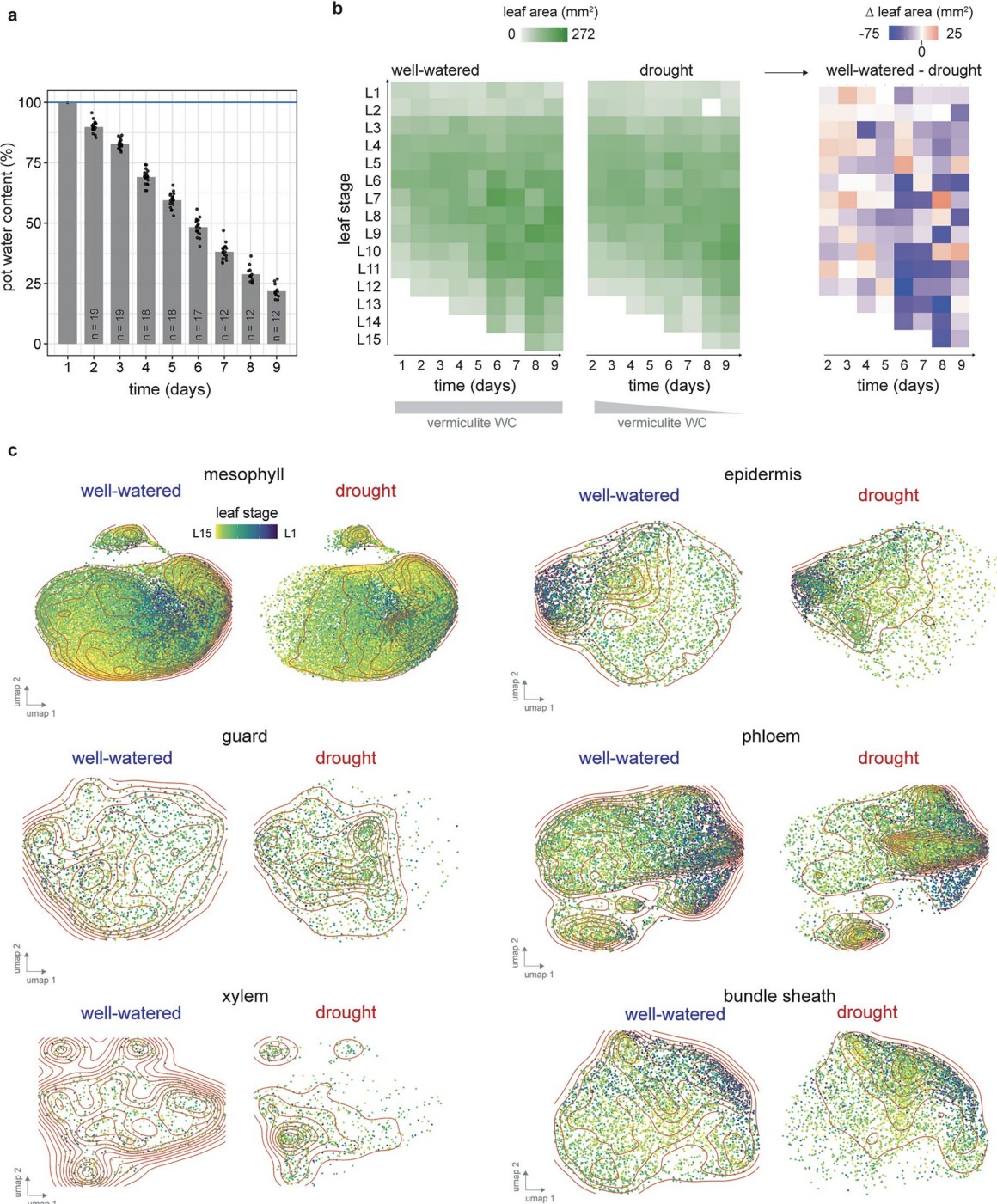

**Extended Data Fig. 5 | Inducing drought stress and measuring its effect on leaf maturation.** (**a**) By withholding water, vermiculite water content in pots containing Arabidopsis plants declined over a 9-day period. Individual points correspond to measurements from single pots (n displayed), with bars showing the sample mean. The blue horizontal line indicates the water content of saturated vermiculite under well-watered conditions. (**b**) Individual leaf area size of rosettes grown under vermiculite conditions. Drought stress had a significant impact on leaf area (3-way ANCOVA p = 1×10-3). The difference in leaf size (Δ mm2) is displayed on the right, where blue and red indicate a reduction or increase in leaf size respectively (**c**) Equal numbers of nuclei from 6 cell-types sourced from well-watered or drought conditions subclustered and colored by their respective leaf stage (red contour lines represent levels of equal point density).

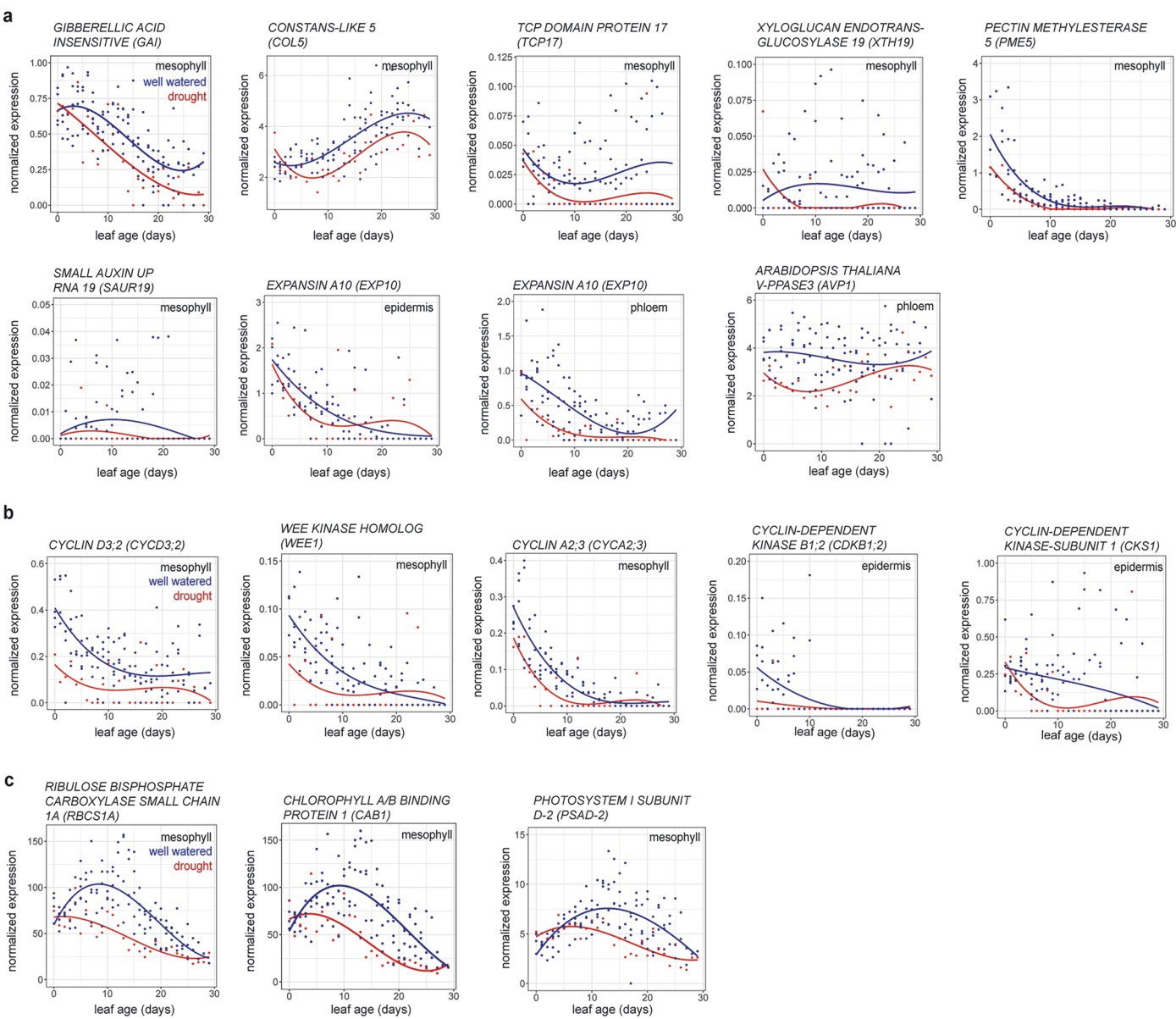

**Extended Data Fig. 6 | Drought stress changes leaf maturation transcriptional dynamics.** Cell type-specific expression patterns of genes involved in either (**a**) leaf development, (**b**) cell cycle, (**c**) photosynthesis, under either well-watered or drought conditions (curve fit using quadratic model).

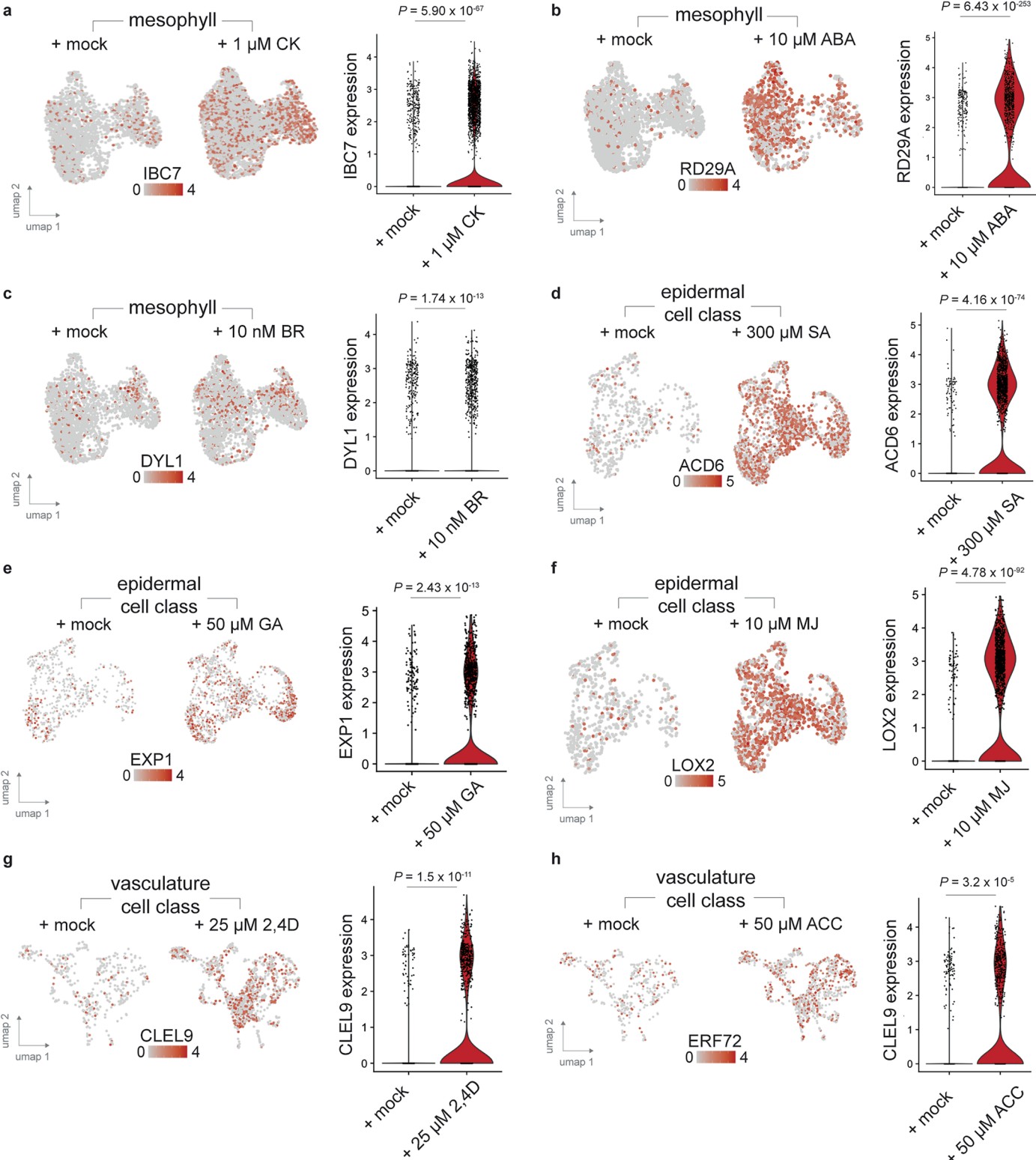

**Extended Data Fig. 7 | Cell-class specific transcriptional responses to exogenous hormone treatment.** UMAPs and accompanying violin plots display a candidate gene's differential expression patterns in response to a hormone treatment within a cell-type class. (**a - c**) The mesophyll cell-type differentially expresses either INDUCED BY CYTOKININ 7 (IBC7), RESPONSE TO DESICCATION 29 A (RD29A), or DORMANCY-ASSOCIATED PROTEIN-LIKE 1 (DYL1) in response to trans-Zeatin (CK), abscisic acid (ABA), or brassinolide (BR) treatment, respectively. (**d - f**) The epidermal cell-class (combining epidermal, guard and trichome cell types) differentially expresses either ACCELERATED CELL DEATH 6 (ACD6), EXPANSIN A1 (EXP1), or LIPOXYGENASE 2 (LOX2) in response to salicylic acid (SA), gibberellin (GA) or methyl jasmonate (MJ), treatment respectively. (**g - h**) The vasculature cell-class (combining phloem, xylem, bundle sheath, hydathode, myrosin idioblast cell types) differentially express either CLE LIKE 9 (CLEL9) or ETHYLENE RESPONSE FACTOR 72 (ERF72) in response to either synthetic auxin (2,4-D) or ethylene precursor 1-Aminocyclopropane-1-carboxylic acid (ACC) respectively. Differential expression was assessed using a likelihood ratio test (adj. p indicated, see Methods). Additional hormone-responsive genes are listed in Supplementary Table 6.

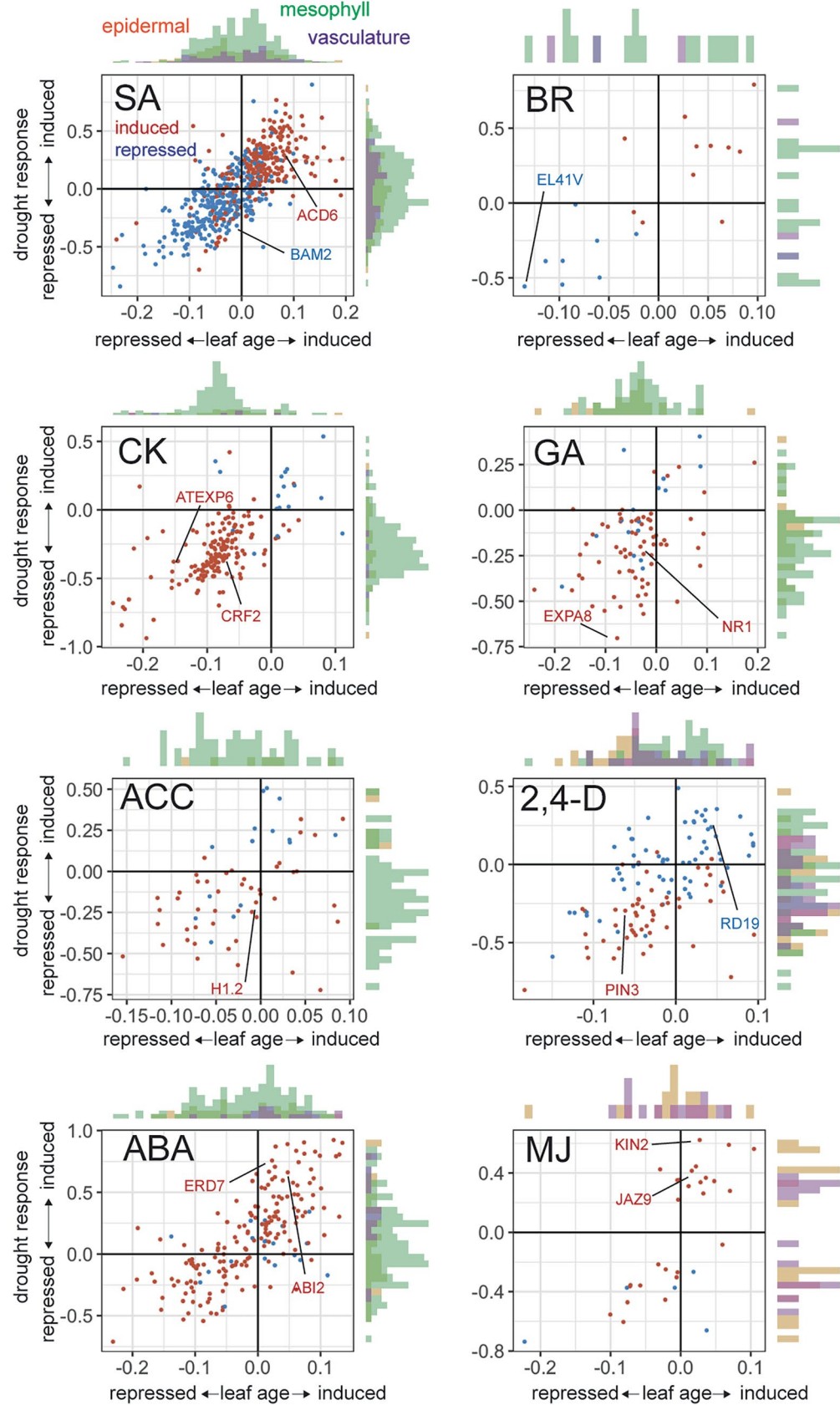

**Extended Data Fig. 8 | Overlapping hormone-responsive genes with those found differentially expressed during leaf maturation or in response to drought stress. Induction (red) or repression (blue) of genes significantly differentially expressed in response to hormone treatment, and each gene's respective induction or repression in response to leaf maturation or drought stress (axis units are coefficients of linear model).** Histograms indicate which cell-type class the hormone response was detected in. Salicylic acid (SA), brassinosteroid (BR), cytokinin (CK), gibberellin (GA), ethylene precursor 1-aminocyclopropane-1-carboxylic acid (ACC), synthetic auxin (2,4-D), abscisic acid (ABA), methyl jasmonate (MJ). Names of example genes responsive to each hormone are indicated (additional genes are listed in Supplementary Table 6).

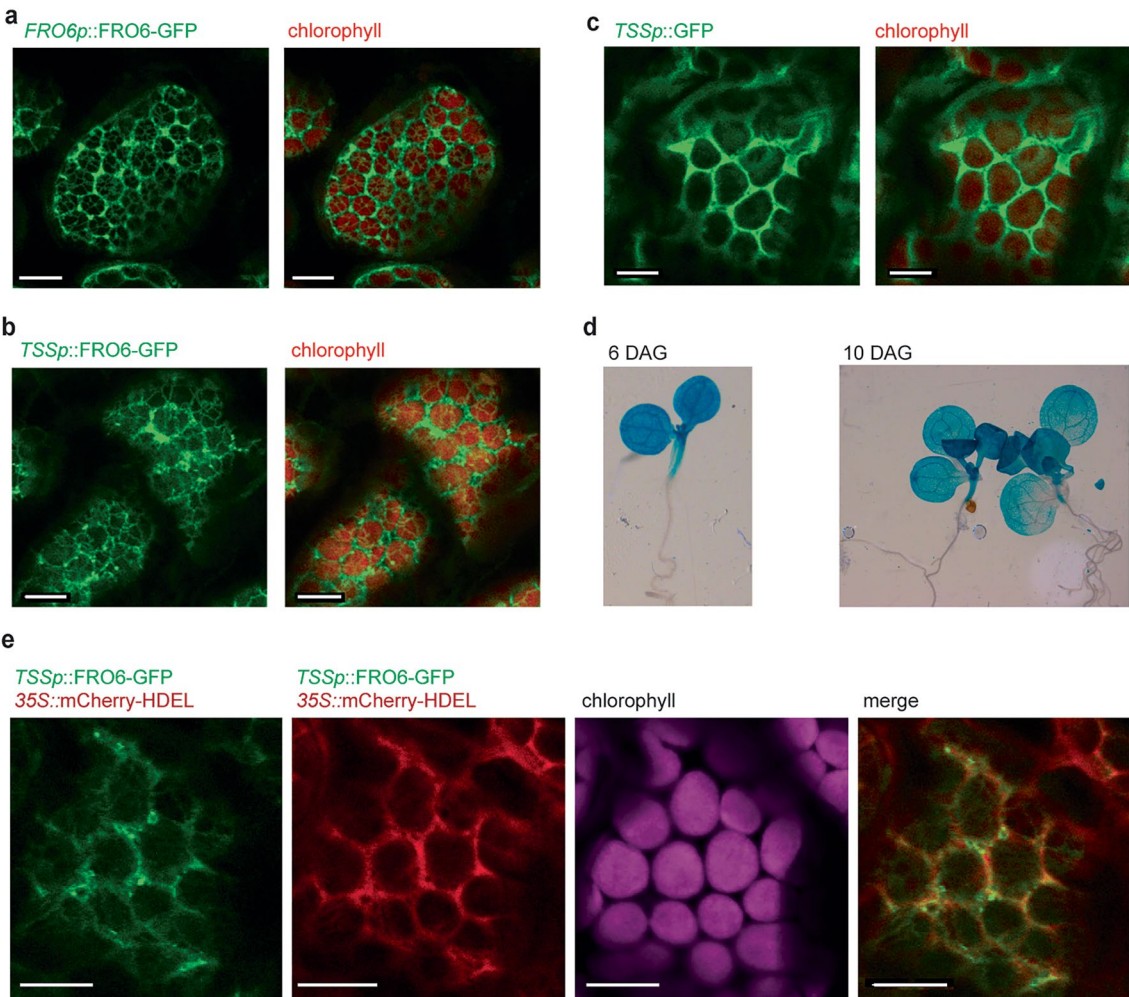

**Extended Data Fig. 9 | Cell-type specific upregulation of FRO6 using the TSSp mesophyll-specific promoter.** (**a**) Confocal microscopy showing FRO6::FRO6-GFP and chlorophyll localization within a mesophyll cell within cotyledons (10μm size indicated). (**b**) Confocal microscopy images showing TSSp::FRO6-GFP (green) and chlorophyll (red) subcellular localization within mesophyll cells (bar represents 10 μm). (**c**) Confocal microscopy images of TSSp::GFP subcellular localization within mesophyll cells (bar represents 10 μm). (**d**) GUS reporter line of TSSp activity in Arabidopsis seedlings 6 and 10 days after sowing (DAS). (**e**) Confocal microscopy images showing FRO6-GFP (green) and mCherry-HDEL (red) localization within mesophyll cells of the TSSp::FRO6-GFP; 35S::mCherry-HDEL genotype (bar represents 10 μm). The final panel shows the overlay of GFP and mCherry signals. Experiments displaying confocal microscopy images were repeated a second independent time and resulted in similar findings.

# Reporting Summary

## Statistics

For all statistical analyses, confirm that the following items are present in the figure legend, table legend, main text, or Methods section.

| n/a | Confirmed | |
|---|---|---|
| ☐ | ☒ | The exact sample size (*n*) for each experimental group/condition, given as a discrete number and unit of measurement |
| ☐ | ☒ | A statement on whether measurements were taken from distinct samples or whether the same sample was measured repeatedly |
| ☐ | ☒ | The statistical test(s) used AND whether they are one- or two-sided<br>*Only common tests should be described solely by name; describe more complex techniques in the Methods section.* |
| ☐ | ☒ | A description of all covariates tested |
| ☐ | ☒ | A description of any assumptions or corrections, such as tests of normality and adjustment for multiple comparisons |
| ☐ | ☒ | A full description of the statistical parameters including central tendency (e.g. means) or other basic estimates (e.g. regression coefficient) AND variation (e.g. standard deviation) or associated estimates of uncertainty (e.g. confidence intervals) |
| ☐ | ☒ | For null hypothesis testing, the test statistic (e.g. *F*, *t*, *r*) with confidence intervals, effect sizes, degrees of freedom and *P* value noted<br>*Give P values as exact values whenever suitable.* |
| ☒ | ☐ | For Bayesian analysis, information on the choice of priors and Markov chain Monte Carlo settings |
| ☒ | ☐ | For hierarchical and complex designs, identification of the appropriate level for tests and full reporting of outcomes |
| ☐ | ☒ | Estimates of effect sizes (e.g. Cohen's *d*, Pearson's *r*), indicating how they were calculated |

*Our web collection on statistics for biologists contains articles on many of the points above.*

## Software and code

Policy information about availability of computer code

| Data collection | no software was used for data collection |
|---|---|
| Data analysis | The following R packages were used: Seurat (v5.2.1), DESeq2 (v1.46.0), ggplot2 (v4.0.0), EDASeq (v2.40.0), dplyr (v1.1.4), data.table (v1.17.0), plyranges (v1.26.0), cowplot (v1.2.0), viridis (v0.6.5), VennDiagram (v1.7.3), rtracklayer (v1.66.0), plyr (v1.8.9), reshape2 (v1.4.4), gridExtra (v2.3), gplots (v3.2.0), plotly (v4.10.4), splines (v4.4.3), igraph (v2.1.4), Matrix (v1.7.3), devtools (v2.4.5), and stringr (v1.5.1).<br>R code is deposited on GitHub at https://github.com/joey1463/Leaf-by-Leaf-Drought-Atlas.git |

For manuscripts utilizing custom algorithms or software that are central to the research but not yet described in published literature, software must be made available to editors and reviewers. We strongly encourage code deposition in a community repository (e.g. GitHub). See the Nature Portfolio guidelines for submitting code & software for further information.

## Data

Policy information about availability of data

All manuscripts must include a data availability statement. This statement should provide the following information, where applicable:
- Accession codes, unique identifiers, or web links for publicly available datasets
- A description of any restrictions on data availability
- For clinical datasets or third party data, please ensure that the statement adheres to our policy

Single-cell RNA-seq data are publicly available through GEO (GSE290214). Nuclei data are hosted at the Klarman Cell Observatory at: https://singlecell.broadinstitute.org/single_cell/study/SCP2703). Microscopy data reported in this paper will be shared by the lead contact upon request.

## Research involving human participants, their data, or biological material

Policy information about studies with human participants or human data. See also policy information about sex, gender (identity/presentation), and sexual orientation and race, ethnicity and racism.

| Reporting on sex and gender | not applicable |
|---|---|
| Reporting on race, ethnicity, or other socially relevant groupings | not applicable |
| Population characteristics | not applicable |
| Recruitment | not applicable |
| Ethics oversight | not applicable |

Note that full information on the approval of the study protocol must also be provided in the manuscript.

# Field-specific reporting

Please select the one below that is the best fit for your research. If you are not sure, read the appropriate sections before making your selection.

☒ Life sciences     ☐ Behavioural & social sciences     ☐ Ecological, evolutionary & environmental sciences

For a reference copy of the document with all sections, see nature.com/documents/nr-reporting-summary-flat.pdf

# Life sciences study design

All studies must disclose on these points even when the disclosure is negative.

| Sample size | For our individual leaf drought time course we chose to harvest and analyze 6 plants per time point (9 time points total). We selected our time points through previous experimental evidence based off bulk transcriptomics that indicated this was the sufficient number of samples to detect gene expression differences.<br>For our individual seedling time course experiment we chose to harvest and analyze 6 plants per time point, per treatment condition (11 time points total). We selected our time points through previous experimental evidence based off bulk transcriptomics that indicated this was the sufficient number of samples to detect gene expression differences. |
|---|---|
| Data exclusions | No significant data was excluded from our study. Only genes with low read counts, or nuclei with low UMIs, were removed during normal processing of single-nuclei sequencing data. |
| Replication | Single nuclei sequencing data presented in this study was replicated in the following ways:<br>- For our individual leaf transcriptome sequencing experiment, we assayed 3 individual plants. We repeated this experiment in a second true biological replicate within a separate time course (6 individual plants total), which yielded comparable results.<br>- For our individual seedling transcriptome sequencing experiment, we assayed 6 - 12 individual plants selected across 3 or more separate agar plates.<br>For our FRO6 transgenic growth assays, we replicated each test on 2 or greater times - all attempts at replication were successful. |
| Randomization | Plants (both whole rosettes and seedlings) were randomized every second or third day within the growth chamber. |
| Blinding | Blinding was not possible because experimental groups were defined by visible phenotypic traits and treatment conditions that could not be concealed during data collection |

# Reporting for specific materials, systems and methods

We require information from authors about some types of materials, experimental systems and methods used in many studies. Here, indicate whether each material, system or method listed is relevant to your study. If you are not sure if a list item applies to your research, read the appropriate section before selecting a response.

## Materials & experimental systems

| n/a | Involved in the study |
|---|---|
| ☒ | ☐ Antibodies |
| ☒ | ☐ Eukaryotic cell lines |
| ☒ | ☐ Palaeontology and archaeology |
| ☒ | ☐ Animals and other organisms |
| ☒ | ☐ Clinical data |
| ☒ | ☐ Dual use research of concern |
| ☐ | ☒ Plants |

## Methods

| n/a | Involved in the study |
|---|---|
| ☒ | ☐ ChIP-seq |
| ☒ | ☐ Flow cytometry |
| ☒ | ☐ MRI-based neuroimaging |

## Dual use research of concern

Policy information about dual use research of concern

### Hazards

Could the accidental, deliberate or reckless misuse of agents or technologies generated in the work, or the application of information presented in the manuscript, pose a threat to:

| No | Yes | |
|---|---|---|
| ☒ | ☐ | Public health |
| ☒ | ☐ | National security |
| ☒ | ☐ | Crops and/or livestock |
| ☒ | ☐ | Ecosystems |
| ☒ | ☐ | Any other significant area |

### Experiments of concern

Does the work involve any of these experiments of concern:

| No | Yes | |
|---|---|---|
| ☒ | ☐ | Demonstrate how to render a vaccine ineffective |
| ☒ | ☐ | Confer resistance to therapeutically useful antibiotics or antiviral agents |
| ☒ | ☐ | Enhance the virulence of a pathogen or render a nonpathogen virulent |
| ☒ | ☐ | Increase transmissibility of a pathogen |
| ☒ | ☐ | Alter the host range of a pathogen |
| ☒ | ☐ | Enable evasion of diagnostic/detection modalities |
| ☒ | ☐ | Enable the weaponization of a biological agent or toxin |
| ☒ | ☐ | Any other potentially harmful combination of experiments and agents |

# Plants

| | |
|---|---|
| Seed stocks | We used wild type Arabidopsis, as well as generated 2 transgenic lines overexpression the FRO6 gene. |
| Novel plant genotypes | A 4198 bp genomic fragment from the FRO6 locus, which contains the FRO6 coding region, 886 bp upstream of ATG, and 211 bp of 3' UTR, was used to generate FRO6p::FRO6:GFP transgenic line.  GFP was translationally fused to the C-terminus of FRO6 before the stop codon. For the TSSp::FRO6:GFP transgenic line, a 1643 bp fragment upstream of TSS start codon was used to replace the FRO6 promoter. The same TSSp fragment was used in generating the TSSp::GUS transcriptional fusion.  All transgenes were cloned into the binary vector pMX202 and stably transformed into Arabidopsis Col-0 background. |
| Authentication | The presence of the transgene was confirmed by growth on selection media (kanamycin), then verified by visualizing GFP expression. 20 positive lines met this criteria. Single inserts were chosen from these lines, and then homozygosed. |

