## [Peer Review File · Nature Plants]

Stress drives plasticity in leaf aging transcriptional dynamics in *Arabidopsis thaliana*

Corresponding Author: Professor Joseph Ecker

Version 0:

Decision Letter:

12th October 2025

Dear Professor Ecker,

Please accept our sincere apologies for the delayed feedback because we had some difficulties to get a referee's report. Your Article, "Stress drives plasticity in leaf aging transcriptional dynamics" has now been seen by 2 referees. You will see from their comments below that while they find your work of interest, some important points are raised. We are very interested in the possibility of publishing your study in Nature Plants, but would like to consider your response to these concerns in the form of a revised manuscript before we make a final decision on publication.

Please highlight all changes in the manuscript text file.

When revising your manuscript please:

- * Include a "Response to referees" document detailing, point-by-point, how you addressed each referee comment. If no action was taken to address a point, you must provide a compelling argument. This response will be sent back to the referees along with the revised manuscript.
- * If you have not done so already please begin to revise your manuscript so that it conforms to our Article format instructions at <http://www.nature.com/nplants/info/final-submission>. Refer also to any guidelines provided in this letter.
- * Include a revised version of any required reporting checklist. It will be available to referees (and, potentially, statisticians) to aid in their evaluation if the manuscript goes back for peer review. A revised checklist is essential for re-review of the paper.
- * Pay close attention to our [href="https://www.nature.com/nature-portfolio/editorial-policies/image-integrity">Digital Image Integrity Guidelines](https://www.nature.com/nature-portfolio/editorial-policies/image-integrity) and to the following points. Please ensure:
 - that unprocessed scans are clearly labelled and match the gels and western blots presented in figures.
 - that control panels for gels and western blots are appropriately described as loading on sample processing controls
 - that all images in the paper are checked for duplication of panels and for splicing of gel lanes.
 - that you retain unprocessed data and metadata files after publication, ideally archiving data in perpetuity. These may be requested during the peer review and production process or after publication if any issues arise.

EXTENDED DATA FIGURES

Nature Plants strongly supports public availability of data and are therefore keen that the data used in your paper is placed in an appropriate public data repository. Alternatively, if this is not possible, you may present the data as Extended Data or Supplementary Information. If data can only be shared on request, please explain why in your Data Availability Statement, and also in the correspondence with your editor. Please note that for some data types, deposition in a public repository is mandatory.

Link Redacted

We hope to receive your revised manuscript within eight weeks. If you cannot send it within this time, please let us know. We will be happy to consider your revision so long as nothing similar has been accepted for publication at Nature Plants or published elsewhere.

Sincerely,

Jun Lyu, PhD
Senior Editor, Nature Plants
<http://www.nature.com/nplants/>
4210-11,42F, The Center, 989 Changle Road
Shanghai, 200031, China

Nature Plants is committed to improving transparency in authorship. As part of our efforts in this direction, we are now requesting that all authors identified as 'corresponding author' on published papers create and link their Open Researcher and Contributor Identifier (ORCID) with their account on the Manuscript Tracking System (MTS), prior to acceptance. This applies to primary research papers only. ORCID helps the scientific community achieve unambiguous attribution of all scholarly contributions. You can create and link your ORCID from the home page of the MTS by clicking on 'Modify my Springer Nature account'. For more information please visit www.springernature.com/orcid.

Reviewers' Comments:

Reviewer #1 (Comments for the Author):

In this manuscript, the authors characterize the cell-level dynamics of leaf growth, maturation and aging, as well as the response of leaves to drought stress. To do this, they leverage combinatorial indexing to sequence a tremendous amount of single nuclei data from leaves, leveraging the nature of combinatorial indexing being traceable to track which cells belong to which leaves. With this large dataset, they identify genes that are specifically up or down regulated across aging within epidermis, while remaining flat in other cell types. Extending this approach to all cell types, they identify genes that are associated both with leaf developmental stage and with time. They then further built upon this atlas of leaf development, sequencing nuclei from leaves under drought stress on *vermiculate*, under various hormonal treatments, and under finely controlled drought stress using hard agar. Finally, they identify a candidate gene whose cell type specific expression was responsible for leaf size, and whose repression was important for the plant's response to drought. They identify FRO6, and test if restoring its expression in the mesophyll will restore leaf size in drought stressed plants. They conclude that its expression with a TSS promoter increases leaf size under drought size but not in well-watered conditions. Overall, this is an outstanding piece of work.

The sole major difficulty in this paper is in Figure 5, with their claim that "Notably, the leaf area and shoot biomass of the transgenic lines did not increase under well-watered conditions." While this is technically true, the claim is misleading. First, their claim that FRO6 plants do not show larger leaf area under well-watered conditions seems mostly supported by the high variance of the samples and the two WT outliers with high area. Without these outliers, or with less variance in the samples (like in the drought stressed samples), it is likely that there would be a significant increase in rosette area and dry weight in the FRO6 expressing well-watered samples. Second, their second allele of FRO6 expressing does not show significant dry weight, area, or shoot increases under droughted conditions ($P = .09$), weakening their claim about FRO6. Third, despite putting an NS for not significant on the well-watered samples, they show their P-value on non-significant samples that are droughted. This should instead be consistent. Because of these issues, the section on the importance of FRO6 will either need to be better supported or the claims removed.

Besides this major issue in the paper, there are several small issues that should be addressed. Their computational methods need more details on how commands were run in order to ensure replicability and evaluate results. For example, they used `FindMarkers()` to identify differentially expressed genes following hormone treatment but do not mention what parameters they sent when running the command. Additionally, when running the command they take genes with adjusted P-values of below .1, instead of .05. This choice should be explained. It would also be helpful for the leaves in Figure 1A to be labeled similarly to Figure 2A. Minorly, line 402 has an extra space before the citation.

Outside of these major issues with FRO6 in Figure 5, I believe the manuscript represents an essential resource for the community, and that the data analysis identifying aging dynamics in the leaf, the main aspect of the paper, is done well.

Although there are major issues with Figure 5, these are not vital to the main claims of the paper. After being corrected, this

manuscript will make an important contribution to the field by providing a tremendous data resource to the community.

Reviewer #2 (Comments for the Author):

In this manuscript single-nucleus transcriptome analyses were performed to identify Arabidopsis genes which are involved in the restriction of leaf expansion during drought which is a well-known stress-avoidance response. The first finding is that this response involves advances in transcriptional programs associated with leaf aging (premature induction of the programs). The second important finding is that decrease in expression of the FERRIC REDUCTION OXIDASE 6 (FRO6) gene during drought correlates with reduced shoot size. By overexpression of FRO6 under control of a mesophyll specific promoter the authors were able to increase shoot size under drought conditions.

Title: It should be added that the investigation has been done with Arabidopsis. It is meanwhile obvious that Arabidopsis is not a model for all plant species.

Abstract, line 29: "Our findings reveal how the environment can reshape..." I do not agree with this wording. The study has convincingly shown that stress such as drought has impact on the development of leaves by advancing the development/leaf biological age ahead of the "calendar age". The sentence should be revised by specifying the result: "The finding of this study reveal that the environment can reshape developmental trajectories at the cellular level by advancing development.....". In line 30 it is written about "trajectories". This term is used only once in the entire manuscript, is not explained and hence should be replaced.

Throughout this manuscript, the terms senescence, maturation, aging seem to be used as synonymous terms. The processes are however not identical. Several reviews deal with the difference between aging and senescence, e.g. Thomas (2013, New Phytologist 197:696-711). A typical feature of senescence is a decline in chlorophyll content of leaves. To avoid confusion between aging and senescence greenness of leaves would be a useful parameter and could be easily determined by chlorophyll content or light transmission.

Lines 277-279 and Fig. 5c: The subcellular localization of FRO6 was examined by use of a fluorescent reporter line of Arabidopsis. From the microscopy image shown in Fig. 5c it has been concluded that FRO6 appeared to be localized to the "endoplasmatic reticulum and near chloroplasts within mesophyll cells". This precise information is surprising considering that the complete cytosol except chloroplasts shows fluorescence. Either the text has to be rephrased or better images need to be provided. Image b in Extended Data Fig. 9 might seemingly show fluorescence of the endoplasmatic reticulum. However, to support this interpretation a co-localization of FRO6 with a marker of the endoplasmatic reticulum would be required. To show a localization close to chloroplasts, electron microscopy combined with immunogold labelling would be necessary. Extended data Fig 9. It would be important to demonstrate the mesophyll specific expression of the the TSSp::GFP construct by showing complete leaf sections.

Discussion, lines 300-328:

The results obtained by this study indicate that the mesophyll is important for perception of stress (line 319). In my opinion this finding suggests that stress perception requires the presence of mature chloroplasts and/or photosynthesis – typical features of the mesophyll. The authors should briefly discuss the role of chloroplasts as sensors of environment as outlined in many articles on retrograde signaling, e.g. Pfannschmidt et al. 2020 (Phil. Trans. Royal Soc. 375:20190396).

Unfortunately, the molecular reasons underlying this reshaping of leaf development in response to environment have only been discussed briefly. Likely epigenetic changes take place in response to environment, as indicated in line 307.

It is compelling that overexpression of a single gene, i.e. FRO6, has a positive effect on leaf area during stress. Scientifically, it would be interesting to elucidate, how FRO6 exerts its impacts on leaf expansion. Certainly, this is not the question of this study. Nevertheless, it would be important to raise this question in the discussion. And, is there anything known about epigenetic effects of FRO6?

Line 347: "field capacity (FC)" should be explained

Line 620: Here it is written that "N.I.E. performed physiological measurements". Surprisingly, I have not detected any physiological data in this manuscript. Leaves were just characterized by morphological data (area and dry weight, Fig. 5) in addition to gene expression.

Language:

For me it is sometimes hard to understand the sentences. Language certainly needs to be concise, but here sometimes the contraction of language has been done at the expense of exactness. E.g. the title should be more precise, i.e. "Stress drives plasticity in transcriptional dynamics during leaf aging in Arabidopsis", see also comment above concerning the title.

Jargon used in laboratories is not appropriate in a scientific manuscript. E.g. line 275: "vermiculite experiment". Another example is "vermiculite drying" in line 225.

In line 408 it is written "where we overlapped the bundle-..." I guess, this is also lab jargon.

*****END*****

Version 1:

Decision Letter:

Our ref: NPLANTS-250720382A

19th January 2026

Dear Dr. Ecker,

Thank you for submitting your revised manuscript "Stress drives plasticity in leaf aging transcriptional dynamics in *Arabidopsis thaliana*" (NPLANTS-250720382A). It has now been seen by one of the original referees and the comment is below. We'll be happy in principle to publish it in Nature Plants, pending minor revisions to comply with our editorial and formatting guidelines.

As you can see below, Referee #1 is now happy with the manuscript. Referee #2 unfortunately couldn't review the manuscript this time, but we read and discussed your response to the concerns and we think you addressed them enough for publication.

At this stage, given that there is no more concern from referees, you can just read your manuscript one last time to be sure you like every word of it, before it is published for eternity. While you are working on this, we will be performing detailed checks on your paper and will send you a checklist detailing our editorial and formatting requirements in about a week (hopefully). Please do not upload the final materials until you receive this additional information from us. Once you receive the checklist, you can integrate all changes into a final version.

==> IMPORTANT: As the current version of your manuscript is in a PDF format, please email us *now* a copy of a text-only file in an editable format (Microsoft Word or LaTeX) as we cannot proceed with PDFs. Don't forget to CC plants@nature.com.

Thank you again for your interest in Nature Plants. Please do not hesitate to contact me if you have any questions.

Sincerely,

--
Guillaume Tena
Editor
Nature Plants

***** Reviewer Reports *****

--Reviewer #1:

I thank the authors for their consideration of my remarks and am satisfied with the changes made.

***** END *****

Version 2:

Decision Letter:

23rd February 2026

Dear Professor Ecker,

We are pleased to inform you that your Article entitled "Stress drives plasticity in leaf aging transcriptional dynamics in *Arabidopsis thaliana*", has now been accepted for publication in Nature Plants.

Over the next few weeks, your paper will be copyedited to ensure that it conforms to Nature Plants style. We look particularly carefully at the titles of all papers to ensure that they are relatively brief and understandable.

Once your paper is typeset, you will receive an email with a link to choose the appropriate publishing options for your paper and our Author Services team will be in touch regarding any additional information that may be required.

Acceptance of your manuscript is conditional on all authors' agreement with our publication policies (see <http://www.nature.com/authors/policies/index.html>). In particular your manuscript must not be published elsewhere.

Authors may need to take specific actions to achieve compliance with funder and institutional open access

mandates. If your research is supported by a funder that requires immediate open access (e.g. according to [Plan S principles](https://www.springernature.com/gp/open-science/plan-s-compliance) or the [NIH public access policy](https://www.springernature.com/gp/open-science/us-federal-agency-compliance)) then you should select the gold OA route, and we will direct you to the compliant route where possible. Because authors warrant under our subscription licensing terms that they haven't committed to licensing any version of their article under a licence inconsistent with the terms of our agreement – including the applicable embargo period – publication under the subscription model isn't suitable for authors whose funders require no embargo.

We welcome the submission of potential cover material (including a short caption of around 40 words) related to your manuscript; suggestions should be sent to Nature Plants as electronic files (the image should be 300 dpi at 210 x 297 mm in either TIFF or JPEG format). Please note that such pictures should be selected more for their aesthetic appeal than for their scientific content, and that colour images work better than black and white or grayscale images. Please include a written description of your image and how it was created along with your image files. Please do not try to design a cover with the Nature Plants logo etc., and please do not submit composites of images related to your work. I am sure you will understand that we cannot make any promise as to whether any of your suggestions might be selected for the cover of the journal.

With kind regards,

Raphael Trösch
Editor
Nature Plants

P.S. Click on the following link if you would like to recommend Nature Plants to your librarian
<http://www.nature.com/subscriptions/recommend.html#forms>

** Visit the Springer Nature Editorial and Publishing website at http://editorial-jobs.springernature.com?utm_source=ejP_NPlan_email&utm_medium=ejP_NPlan_email&utm_campaign=ejp_NPlan for more information about our career opportunities. If you have any questions please click [here](mailto:editorial.publishing.jobs@springernature.com).

Reviewer #1 (Comments for the Author):

In this manuscript, the authors characterize the cell-level dynamics of leaf growth, maturation and aging, as well as the response of leaves to drought stress. To do this, they leverage combinatorial indexing to sequence a tremendous amount of single nuclei data from leaves, leveraging the nature of combinatorial indexing being traceable to track which cells belong to which leaves. With this large dataset, they identify genes that are specifically up or down regulated across aging within epidermis, while remaining flat in other cell types. Extending this approach to all cell types, they identify genes that are associated both with leaf developmental stage and with time. They then further built upon this atlas of leaf development, sequencing nuclei from leaves under drought stress on vermiculate, under various hormonal treatments, and under finely controlled drought stress using hard agar. Finally, they identify a candidate gene whose cell type specific expression was responsible for leaf size, and whose repression was important for the plant's response to drought. They identify FRO6, and test if restoring its expression in the mesophyll will restore leaf size in drought stressed plants. They conclude that its expression with a TSS promoter increases leaf size under drought size but not in well-watered conditions. Overall, this is an outstanding piece of work.

- 1. The sole major difficulty in this paper is in Figure 5, with their claim that “Notably, the leaf area and shoot biomass of the transgenic lines did not increase under well-watered conditions.” While this is technically true, the claim is misleading. First, their claim that FRO6 plants do not show larger leaf area under well-watered conditions seems mostly supported by the high variance of the samples and the two WT outliers with high area. Without these outliers, or with less variance in the samples (like in the drought stressed samples), it is likely that there would be a significant increase in rosette area and dry weight in the FRO6 expressing well-watered samples.**

We thank the reviewer for bringing this point to our attention. Within our revised manuscript, we have now included another drought assay testing the growth responses of Col-0 and the 2 TSSp::FRO6 alleles on soil. This experiment included a higher number of individual plants (24 per treatment, per genotype); thus providing greater statistical power. Within this experiment, we found that both transgenic alleles displayed higher shoot biomass under drought conditions ($p < 0.05$). Moreover, under well-watered conditions, a statistically significant difference was not observed ($p > 0.1$). The results of this experiment are presented in new **Figure 5f**.

- 2. Second, their second allele of FRO6 expressing does not show significant dry weight, area, or shoot increases under droughted conditions ($P = .09$), weakening their claim about FRO6.**

To address this concern, we have now updated our text to distinguish between what is significant ($p < 0.05$), marginally significant ($0.05 < p < 0.1$) and non-significant ($p > 0.1$). Moreover, we specifically state that allele 2 was the weaker performing allele in this regard. Specifically, on lines 284-188, we write:

Similarly, when we grew these transgenic lines on vermiculite, we observed increased shoot biomass and leaf area under drought conditions (Fig. 5d,g,h), with TSSp::FRO6-GFP allele 1 significantly different ($p < 0.05$) and allele 2 marginally significant ($0.05 < p < 0.1$). This trend persisted when we examined the shoot size of transgenic lines grown on HA stress (Fig. 5i). Notably, the shoot biomass and leaf area of the transgenic lines did not increase significantly under non-stress conditions (Fig. 5f-i, $p > 0.1$).

- 3. Third, despite putting an NS for not significant on the well-watered samples, they show their P-value on non-significant samples that are droughted. This should instead be consistent. Because of these issues, the section on the importance of FRO6 will either need to be better supported or the claims removed.**

We thank the reviewer for this comment. In our newly updated Figure 5, we report all p-values, regardless of their significance, and no longer use the “NS” designation.

- 4. Besides this major issue in the paper, there are several small issues that should be addressed. Their computational methods need more details on how commands were run in order to ensure replicability and evaluate results. For example, they used FindMarkers() to identify differentially expressed genes following hormone treatment but do not mention what parameters they sent when running the command.**

To address this concern, we have now published the R scripts used for single cell analyses on GitHub (available here - <https://github.com/joey1463/Leaf-by-Leaf-Drought-Atlas/tree/main>). This repository describes all parameters used within our analysis. Similarly, we have now included a Data Availability section to the manuscript (lines 499-502) that refers to this link.

5. **Additionally, when running the command they take genes with adjusted P-values of below .1, instead of .05. This choice should be explained.**

We have amended our analysis to include hormone responsive genes with an FDR corrected p-value of 0.05, instead of 0.1. Thus, **Figure 3a, 3b, 3d** and **Extended Data Fig. 8** have been updated to reflect this change. This does not impact the main findings of this analysis. We thank the reviewer for this comment.

6. **It would also be helpful for the leaves in Figure 1A to be labeled similarly to Figure 2A.**

Our new **Figure 1a** reflects this requested change. Additionally, to help ensure the reader understands how the atlas reveals changes in gene expression with leaf stage and time, a new panel **1c** is included. The gene expression plot that this new panel replaces can be found in **Extended Data Figure 3**.

7. **Minorly, line 402 has an extra space before the citation.**

Thank you, this change has been made.

8. **Outside of these major issues with FRO6 in Figure 5, I believe the manuscript represents an essential resource for the community, and that the data analysis identifying aging dynamics in the leaf, the main aspect of the paper, is done well. Although there are major issues with Figure 5, these are not vital to the main claims of the paper. After being corrected, this manuscript will make an important contribution to the field by providing a tremendous data resource to the community.**

We thank the reviewer for the positive feedback.

Reviewer #2 (Comments for the Author):

In this manuscript single-nucleus transcriptome analyses were performed to identify *Arabidopsis* genes which are involved in the restriction of leaf expansion during drought which is a well-known stress-avoidance response. The first finding is that this response involves advances in transcriptional programs associated with leaf aging (premature induction of the programs). The second important finding is that decrease in expression of the FERRIC REDUCTION OXIDASE 6 (FRO6) gene during drought correlates with reduced shoot size. By overexpression of FRO6 under control of a mesophyll specific promoter the authors were able to increase shoot size under drought conditions.

1. **Title: It should be added that the investigation has been done with *Arabidopsis*. It is meanwhile obvious that *Arabidopsis* is not a model for all plant species.**

We agree with the reviewer, and have changed the title to “Stress drives plasticity in leaf aging transcriptional dynamics in *Arabidopsis thaliana*”.

2. **Abstract, line 29: “Our findings reveal how the environment can reshape...”. I do not agree with this wording. The study has convincingly shown that stress such as drought has impact on the development of leaves by advancing the development/leaf biological age ahead of the “calendar age”. The sentence should be revised by specifying the result: “The finding of this study reveal that the environment can reshape developmental trajectories at the cellular level by advancing development.....”.**

We agree with the reviewer and have made this sentence specifically refer to how gene expression can advance in response to drought. This new sentence reads “*Our findings demonstrate how gene expression is reshaped by environmental cues to ensure shoot architecture is adaptive to stress severity.*” (lines 30 – 31)

3. **In line 30 it is written about “trajectories”. This term is used only once in the entire manuscript, is not explained and hence should be replaced.**

We agree with the reviewer, and have removed the term ‘trajectories’ from the Abstract.

4. **Throughout this manuscript, the terms senescence, maturation, aging seem to be used as synonymous terms. The processes are however not identical. Several reviews deal with the difference between aging and senescence, e.g. Thomas (2013, New Phytologist 197:696-711). A typical feature of senescence is a decline in chlorophyll content of leaves. To avoid confusion between aging and senescence greenness of leaves would be a useful parameter and could be easily determined by chlorophyll content or light transmission.**

We thank the reviewer for raising this concern. Indeed, in our initial manuscript, these terms were used somewhat interchangeably. Within our revised manuscript, we have now defined what we consider ‘aging’, as guided by the article the reviewer kindly provided. Specifically, on lines 83-84, we now define aging as:

Leaf aging is defined by the progression of a leaf’s physiology over time, from its initial emergence through to the last stages of its lifespan⁹.

Having defined ‘age’ in this way, we no longer use it interchangeably with ‘maturation’ and ‘senescence’ within the text, and have made these substitutions accordingly. Indeed, we now only refer to senescence when we describe genes that have prior literature associating it with this process.

5. **Lines 277-279 and Fig. 5c: The subcellular localization of FRO6 was examined by use of a fluorescent reporter line of Arabidopsis. From the microscopy image shown in Fig. 5c it has been concluded that FRO6 appeared to be localized to the “endoplasmic reticulum and near chloroplasts within mesophyll cells”. This precise information is surprising considering that the complete cytosol except chloroplasts shows fluorescence. Either the text has to be rephrased or better images need to be provided. Image b in Extended Data Fig. 9 might seemingly show fluorescence of the endoplasmic reticulum. However, to support this interpretation a co-localization of FRO6 with a marker of the endoplasmic reticulum would be required. To show a localization close to chloroplasts, electron microscopy combined with immunogold labelling would be necessary.**

To address the reviewer’s concern about ER-localization, we crossed our *TSSp::FRO6-GFP* transgenic line with a *35S::mCherry-HDEL* transgenic line. The HDEL signal peptide, when present at the C-terminus, localizes to the endoplasmic reticulum, as initially described in *Denecke, J., et al Plant and mammalian sorting signals for protein retention in the endoplasmic reticulum contain a conserved epitope. EMBO J 11, 2345-2355 (1992).*

When we observed GFP and mCherry fluorescence, we found that they are co-localized, indicating that FRO6 is indeed localized to the endoplasmic reticulum. This result is presented in a new panel in **Extended Data Figure 9**. The reviewer’s second concern is whether FRO6-GFP is localized near the chloroplasts in the mesophyll cells. We agree with the reviewer that only EM data can make such determination. Unfortunately, in our hands, the amount of FRO6-GFP fusion was too low for immunoEM detection using an anti-GFP antibody. We therefore removed the description of its localization near the chloroplasts.

6. **Extended data Fig 9. It would be important to demonstrate the mesophyll specific expression of the TSSp::GFP construct by showing complete leaf sections.**

We thank the reviewer for this suggestion. The “mesophyll-specific expression” of TSS was based on the *in situ* hybridization results using an anti-sense TSS probe (Skylar, A, et al. 2011. Metabolic sugar signal promotes Arabidopsis meristematic proliferation via G2. *Developmental biology*, 351(1), 82-89.). We agree with the reviewer that this is insufficient to support the conclusion that the TSS promoter fragment used in our study exclusively drives ectopic mesophyll expression. Moreover, there does appear to be residual expression of TSS within other cell types of the atlas (for example, **Figure 5e**). Unfortunately, we cannot not use *TSSp::GFP* to address the reviewer’s concerns because GFP tends to diffuse between cell layers in younger tissues (for example, see *Wu, X, et al. "Modes of intercellular transcription factor movement in the Arabidopsis apex." (2003): 3735-3745.*). Although FRO6-GFP is ER-localized and could be used for this purpose, its low abundance makes it unsuitable for determining whether there is a small amount of FRO6-GFP in other cell types.

Consequently, we have changed our wording in the manuscript to ensure it is clear that TSS expression is highest within the mesophyll according to our atlas, but is not exclusively expressed there. On lines 274-279 we now state – “Among these, the *TPR-DOMAIN SUPPRESSOR OF STIMPY (TSS, also called REDUCED CHLOROPLAST COVERAGE 2)* stood out for its mesophyll-enriched expression that exceeded FRO6 levels by approximately 2-fold (**Fig. 5e, Extended Data Fig. 9**). We note TSS itself was neither significantly repressed by drought nor by HA

stress (adj. $p > 0.05$, linear model), nor was TSSp active within root tissue (**Extended Data Fig. 9**), however it did display lower expression in other cell types within our atlas (Fig. 5e)."

- 7. Discussion, lines 300-328: The results obtained by this study indicate that the mesophyll is important for perception of stress (line 319). In my opinion this finding suggests that stress perception requires the presence of mature chloroplasts and/or photosynthesis – typical features of the mesophyll. The authors should briefly discuss the role of chloroplasts as sensors of environment as outlined in many articles on retrograde signaling, e.g. Pfannschmidt et al. 2020 (Phil. Trans. Royal Soc. 375:20190396).**

We thank the reviewer for this point. We now include this point in lines 314-318 of the discussion, which includes the suggested reference:

This highlights the mesophyll as an active site of stress signaling during organ development - a role not commonly attributed to this cell type. Further study may reveal the mechanism by which FRO6 can increase shoot size under stress. Similarly, given FRO6's activity in the mesophyll, future work might explore the role that chloroplast signaling plays in stress perception³⁷.

- 8. Unfortunately, the molecular reasons underlying this reshaping of leaf development in response to environment have only been discussed briefly. Likely epigenetic changes take place in response to environment, as indicated in line 307.**

In our revised manuscript, we now build on our description of epigenetic literature that supports our hypothesis. Specifically, on lines 301-304, we now write:

These results agree with discoveries made at the epigenetic level. For example, the advancement of leaf biological age ahead of calendar age was recently captured at the epigenetic level³³. Similarly, evidence points to the stress avoidance response being in part regulated epigenetically³⁴.

Where reference 34 cites - Kooke, R. et al. Epigenetic basis of morphological variation and phenotypic plasticity in *Arabidopsis thaliana*. *Plant Cell* 27, 337-348 (2015).

- 9. It is compelling that overexpression of a single gene, i.e. FRO6, has a positive effect on leaf area during stress. Scientifically, it would be interesting to elucidate, how FRO6 exerts its impacts on leaf expansion. Certainly, this is not the question of this study. Nevertheless, it would be important to raise this question in the discussion. And, is there anything known about epigenetic effects of FRO6?**

We thank the reviewer for this point. We now raise the question regarding the mechanism by which FRO6 increases shoot size under stress in lines 316-318 of the discussion:

Further study may reveal the mechanism by which FRO6 can increase shoot size under stress. Similarly, given FRO6's activity in the mesophyll, future work might explore the role that chloroplast signaling plays in stress perception³⁷.

- 10. Line 347: "field capacity (FC)" should be explained**

We have now explained field capacity in greater detail. Specifically, lines 338-340 now read:

To introduce drought stress, excess water was removed until all pots reached 100% field capacity (FC) – that is, the condition where the vermiculite could hold the maximum amount of water without dripping. We note that in the context of this manuscript, 100% FC is synonymous with 100% water content (WC).

- 11. Line 620: Here it is written that "N.I.E. performed physiological measurements". Surprisingly, I have not detected any physiological data in this manuscript. Leaves were just characterized by morphological data (area and dry weight, Fig. 5) in addition to gene expression.**

We have clarified the author contributions to be more precise about physiological measurements. Specifically we now state on line 645 : "... J.S. and X.W. performed physiological measurements. J.X. and N.I.E. developed and implemented the Shoot Growth Tracker software...."

- 12. Language: For me it is sometimes hard to understand the sentences. Language certainly needs to be concise, but here sometimes the contraction of language has been done at the expense of exactness. E.g.**

the title should be more precise, i.e. “Stress drives plasticity in transcriptional dynamics during leaf aging in *Arabidopsis*”, see also comment above concerning the title.

We thank the reviewer for this feedback. As indicated in a previous comment, we have now adjusted the title of our resubmitted manuscript. Additionally, in our revised manuscript we have adjusted the text to make the wording more precise where we thought it might be unclear. We have also added sentences in areas where we thought the language might have been too contracted (see track changes).

13. Jargon used in laboratories is not appropriate in a scientific manuscript. E.g. line 275: “vermiculite experiment”. Another example is “vermiculite drying” in line 225.

We thank the reviewer for noticing this jargon. We have made the following edits for clarity:

- We have now replaced “vermiculite experiment” in new lines 263-265 ‘Secondly, to independently confirm that a gene was drought responsive, we assessed whether they were differentially expressed in rosette leaves subjected to drying in pots (**Fig. 2a**).’
- We have now replaced “vermiculite drying” in new lines 218-220: “To this end, we exposed *Arabidopsis* seedlings to a range of controlled drought stress levels using the 'hard agar' (HA) system, which can afford more fine-scale adjustment of water potential compared to soil drying.”

14. I line 408 it is written “where we overlapped the bundle-...” I guess, this is also lab jargon.

We have updated the text to provide more clarity. Specifically, on lines 409-412 we now state: “Here, we used the bundle sheath cell type specific markers described in Kim *et. al*⁴⁴ (with an enrichment score > 6) to identify the bundle sheath cell type within our atlas. This was done by assessing how many of these published bundle-sheath marker genes were the same as the cluster specific genes present in our atlas (with an enrichment score >3).”